# EPI-SauriCas9-based mouse ovarian cancer models recapitulating pten deletion in patients
Wutao Chen [1,2,8], Pengju He[3,8], Ling Ding[4,8], Weihua Lou[1,8], Yishu Wang[5], Weiwei Shi[1,2], Zhangzhengyi Fan[3], Yumeng Sheng[3], Jing Luo[3], Zhixing Tan[3], You Wang [1,2,3,9] ✉, Wen Di [1,2,3,9] ✉, Xiaoping Ke [6,9] ✉ & Bin Yu [3,7,9] ✉

Ovarian cancer remains a deadly gynecological malignancy, with *PTEN* loss and *TP53* mutations frequently implicated in its progression. However, suitable models for studying ovarian cancers with *PTEN* and *TP53* deletions are rare. Here we develop and validate the mouse ovarian epithelium with *Pten* and *Trp53* deletions (MEPP) model using the EPI-SauriCas9 system. We demonstrate the role of *Pten* loss in promoting tumorigenicity and metastasis. Single-cell RNA sequencing reveals distinct epithelial subpopulations with varying metastatic potential. MEPP also recapitulates key features of human ovarian cancer, including its immune landscape and therapeutic responses. High-throughput drug screening identifies FK228 and thioguanine as promising therapeutic candidates, both of which show in vivo efficacy and are validated in *PTEN*-deleted organoids. Together, these results establish MEPP as a platform for studying *PTEN*-deleted ovarian cancer and provide a strategy for generating clinically relevant tumor models through targeted gene editing.

Ovarian cancer (OC) represents the most lethal gynecological malignancy in women[1,2]. Despite initial responsiveness to first-line chemotherapy, ~70% of patients experience disease recurrence and develop platinum resistance, necessitating alternative therapeutic strategies[3,4]. Beyond conventional platinum-based chemotherapy and poly (ADP-ribose) polymerase inhibitors (PARPi), novel approaches, including immunotherapies, have demonstrated only modest response rates in OC, largely attributable to its complex mutational landscape[5–8].

PTEN is a critical tumor suppressor and a down-regulator of the PI3K signaling pathway, PTEN dysfunction contributing to OC tumorigenesis[9,10]. Despite that *PTEN* mutations are relatively rare in high-grade serous ovarian carcinoma (HGSOC)[11], loss of *PTEN* expression is frequent in HGSOC and is related to poorer survival[12–14]. Moreover, enhanced PI3K pathway is common in ovarian cancer cells and acted as a valuable drug target[15,16]. This underscores the pressing need to develop more effective therapeutic strategies for ovarian cancers with *PTEN* loss and altered PI3K signaling. Nevertheless, the impact of *PTEN* loss on ovarian cancer tumorigenesis, metastasis, and the tumor microenvironment (TME) remains unclear[17].

Despite the poor prognosis of ovarian cancer, preclinical models that accurately reflect its genetic profile and maintain immunocompetence remain scarce. Human ovarian cancer organoids have transformed research by providing patient-derived 3D cultures that closely mimic tumor histology, genomics, and drug responses. Studies have confirmed their ability to retain primary tumor characteristics, making them reliable for drug evaluation[18–22]. Additionally, organoids derived from both human and mouse oviduct and ovarian surface epithelium (OSE) offered insights into tumorigenesis of ovarian cancers[23–25]. However, limitations persist, including the absence of immune and stromal components[19], challenges in sustaining long-term cultures[20], and poor survival after implantation in immunodeficient mice[26]. Thus, there is an urgent need for models incorporating patient-specific mutations.

[1]Department of Obstetrics and Gynecology, Renji Hospital, School of Medicine, Shanghai Jiao Tong University, Shanghai, China. [2]Shanghai Key Laboratory of Gynecologic Oncology, Renji Hospital, School of Medicine, Shanghai Jiao Tong University, Shanghai, China. [3]State Key Laboratory of Systems Medicine for Cancer, Renji Hospital, School of Medicine, Shanghai Jiao Tong University, Shanghai, China. [4]Traditional Chinese Medicine Department, Renji Hospital, School of Medicine, Shanghai Jiao Tong University, Shanghai, China. [5]Department of Neurology, Renji Hospital, School of Medicine, Shanghai Jiao Tong University, Shanghai, China. [6]Department of Obstetrics and Gynecology, Yangpu Hospital, School of Medicine, Tongji University, Shanghai, China. [7]Shanghai Key Laboratory for Cancer Systems Regulation and Clinical Translation (CSRCT), Shanghai, China. [8]These authors contributed equally: Wutao Chen, Pengju He, Ling Ding, Weihua Lou.[9]These authors jointly supervised this work: You Wang, Wen Di, Xiaoping Ke, Bin Yu. ✉e-mail: wanghh0163@163.com; diwen163@163.com; Xiaoping.Ke@tongji.edu.cn; yubinrenji@outlook.com

Over the past decades, numerous attempts have been made to establish clinically relevant mouse models for ovarian cancer research. The pioneering syngeneic model, the ID8 cell line, developed by Roby et al.[27], has been utilized extensively to investigate immune system dynamics in disease establishment and progression[28,29]. However, in vitro spontaneous transformation resulted in only limited immunogenicity of ID8 and its derivatives following in vivo passages. To address this, Walton's group developed double knockout (Trp53; Brca2) ID8 tumor models applicable to HGSOC research[30], followed by other ID8-based syngeneic models with distinct TME[17,31,32]. Despite these advances, cell line-derived models face genetic instability with prolonged passaging[33–35]. While genetically engineered mouse models (GEMMs) of HGSOC have been investigated, their implementation in research has been hampered by mixed genetic backgrounds and extensive breeding requirements[36,37]. Attempts to generate ovarian cancer models from mouse oviduct or OSE have also been explored but require in vivo electroporation or genetically engineered mice, increasing technical complexity[10,24,38,39]. Thus, there is an urgent need for immunocompetent ovarian cancer models that are genetically stable and easy to establish.

In this study, we present a mouse ovarian cancer cell line, MEPP, generated from mouse ovarian surface epithelium (MOSE) through EPI-SauriCas9-mediated Trp53/Pten double knockout. We demonstrated that MEPP serves as a valuable platform for investigating ovarian cancer in an immunocompetent context, enabling the exploration of TME interactions in tumorigenesis and metastasis, while simultaneously providing a robust platform for therapeutic drug discovery to expand the current OC treatment arsenal. Notably, we further validated MEPP's relevance by matching it with PTEN-deleted human ovarian cancer organoids, providing additional support for the model's clinical applicability and offering a method for establishing tumor models through in vitro gene editing system to simulate other personalized tumor models.

## Results

### Patient ovarian organoids with *PTEN* deletion are capable of long-term propagation

We first examined patient-derived ovarian cancer organoids (PDOs) to motivate our genetic design. Among PDOs that persisted beyond passage 4, most harbored *PTEN* exon-8 variants by sanger sequencing; one line (PO1) expanded for >10 passages (Supplementary Data 1; Supplementary Fig. 1). Single-cell RNA sequencing (scRNA-seq) of PO1 resolved multiple epithelial and stromal clusters with directional transcriptomic concordance to epithelial subsets in human ovarian tumors[40,41], indicating preservation of disease-relevant states (Supplementary Fig. 2). Despite sustained in vitro growth, PO1 did not form subcutaneous tumors in NCG mice at 6 weeks, underscoring the practical limits of PDOs for in vivo studies. Given the frequency of *PTEN* and *TP53* alterations in patients[12,42] and the need for an immunocompetent platform, these observations motivated a mouse model with dual *Pten/Trp53* loss.

### Development and optimization of the EPI-SauriCas9 system for sgRNA screening

To enable simultaneous editing, we engineered an EPI-SauriCas9 dual-sgRNA plasmid that targets *Pten* and *Trp53* in mouse cells (Fig. 1A). Unlike standard transient-expression vectors, the EPI backbone incorporates the EBV OriP origin and the Epstein–Barr nuclear antigen 1 (EBNA1), allowing the plasmid to persist episomally and maintain robust SauriCas9/sgRNA expression. This enables the plasmid vectors to replicate episomally alongside cell division, thereby maintaining plasmid copy numbers during rapid cell proliferation but can also dilute slowly after long-term propagation[43]. The EPI-Cas9 plasmid system has been demonstrated to improve gene editing efficiency in mammalian cells significantly[44,45]. The incorporation of EBNA1 and OriP into the plasmid increased its overall size. To ensure optimal transfection efficiency, we selected the minimal SauriCas9 which has similar efficiency and flexible PAM options compared to SpCas9[46]. And inserted the strong CAG promoter to enhance Cas9 protein

expression. This strategy further elevates Cas9 protein levels in cells with low exogenous DNA expressing capacity, thereby improving gene editing efficiency. This system incorporates the EPI vector for optimal transfection and gene editing, coupled with a dual selection mechanism using Puromycin N-acetyl-transferase and ZsGreen, which allows for both monitoring and selection of successfully transfected cells by puromycin addition or flow cytometry. To test the functionality of this system, mouse hepatocarcinoma cell line Hepa1-6 were transfected with the EPI plasmids containing different sgRNA pairs targeting *Pten* and *Trp53* (Fig. 1B). After transfection, the cells were subjected to puromycin selection and cultured for an additional 12 days to allow plasmid replication and Cas9/sgRNA functioning. Fluorescent microscopy revealed successful ZsGreen expression in the transfected cells, with GFP fluorescence detected as early as 24 h post-transfection (Fig. 1C). Both sgPair1 and sgPair2 produced comparable on-target editing in Hepa1-6 cells (Fig.1D, E). When further tested in the mouse ovarian epithelial line ID8, sgPair1 achieved the highest editing efficiency and was selected for subsequent experiments (Fig. 1F). In addition, we performed sgRNA screening targeting several common mutated genes in ovarian cancer (Supplementary Fig. 3A). Unfortunately, either the sgRNAs exhibited insufficient knockout efficiency, or the MOSE cells failed to survive during long-term passaging after puromycin selection (Supplementary Fig. 3B). These results validate the efficacy and necessity of the EPI-SauriCas9 screening system in facilitating precise genetic alterations in mouse ovarian cancer models.

### EPI-SauriCas9 mediated *Pten* and *Trp53*-deleted mouse ovarian cancer model

Previous findings have demonstrated that *PTEN* deletion facilitated the growth of human ovarian cancer, however, *Pten* loss alone is insufficient for inducing tumorigenesis in mouse ovarian cancer models[47]. Given the high prevalence of *TP53* mutations in ovarian cancer, we sought to develop a mouse model incorporating both *Pten* and *Trp53* deletions. The MOSE was isolated and edited using the EPI-SauriCas9 plasmids with the most efficient sgRNA pair (Fig. 2A, B). Sanger sequencing confirmed successful gene editing of both *Trp53* and *Pten* loci (Fig. 2C). Editing at both *Trp53* and *Pten* was highly efficient, with high indel rates and frameshift mutation rates across early-passage clones (Supplementary Fig. 4A–D). Compared with MOSE, MEPP showed enrichment of cell-cycle/E2F, MYC, and mTORC1 programs, consistent with transcriptional activation following *Trp53/Pten* loss (Supplementary Fig. 4E). Targeted amplicon sequencing at bioinformatically predicted off-target sites for each sgRNA detected no editing above background (Supplementary Fig. 4F). As a negative control for potential EBNA1 effects from the episomal vector, MOSE cells transfected with a scramble sgRNA did not expand beyond passage 3 (Supplementary Fig. 4G). Lentiviral delivery of SauriCas9 and sgRNA pair1 targeting *Trp53* and *Pten* induced efficient editing (Supplementary Fig. 4H), yet deletion of *Trp53* and *Pten* alone did not confer tumorigenicity. Together, these results indicate that EBNA1 expression combined with *Trp53/Pten* loss is required for MOSE transformation. The epithelial origin of MEPP was validated by significantly elevated expression of *Pax8* and *Wt1* compared to mouse adipose mesenchymal stem cells (aMSC) (Fig. 2D).

We assessed the tumorigenic potential of MEPP cells in vivo then. While MOSE cells were unable to form subcutaneous tumors in immunocompetent mice, MEPP cells exhibited robust tumorigenicity and successfully formed subcutaneous tumors (Fig. 2E). Histological analysis of MEPP-derived tumors showed key features of ovarian epithelial tumors, including bulk growth patterns, serous morphology, nuclear pleomorphism, and high WT1 expression, which are consistent with human ovarian cancer (Supplementary Fig. 5A).

To evaluate the metastatic potential of MEPP tumors, we established an orthotopic model, which can better reflect the natural progression of ovarian cancer. In this model, MEPP cells formed orthotopic tumors, induced hemorrhagic ascites, and demonstrated rapid peritoneal dissemination, with metastasis to the omentum, a common site for ovarian cancer metastasis (Fig. 2F). These findings underscore the utility of the

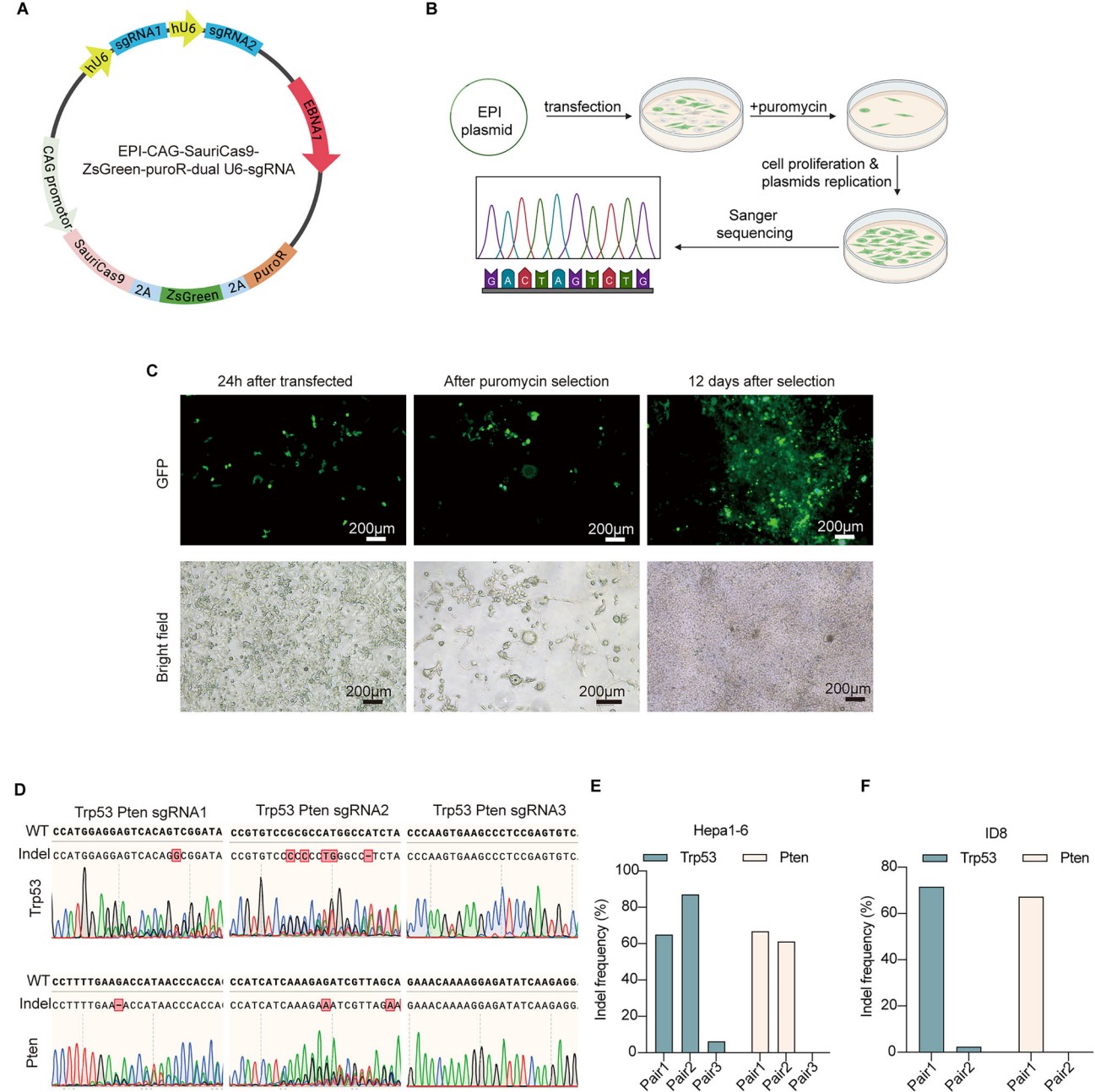

**Fig. 1 | Construction and application of EPI-SauriCas9 screening system.**
**A** Schematic of the EPI-CAG-SauriCas9-ZsGreen-puroR-dual-U6-sgRNA plasmid components. The 2A peptide (T2A or P2A) is incorporated to make polycistron. Created using BioRender.com. **B** Workflow for screening high-efficiency sgRNAs in Hepa1-6 cells via EPI plasmid transfection. GFP is observed 24 h post-transfection. Puromycin selection is applied to enrich successfully transfected cells, followed by 10–12 days of expansion and subsequent sequencing of target loci. Created using BioRender.com. **C** Bright-field microscopy and GFP fluorescence images of Hepa1-6 cells after transfection, selection, and expansion. **D** Knockout efficiency of three sgRNA pairs targeting *Trp53* and *Pten* in Hepa1-6 cells. Sequencing analysis confirms the editing outcomes at the target loci for both genes. Editing frequency of different sgRNA pairs targeting *Trp53* or *Pten* in Hepa1-6 (**E**) and ID8 (**F**).

MEPP model in replicating key aspects of human ovarian cancer, including tumor growth, metastasis, and histopathological features, making it a valuable tool for further mechanistic studies and therapeutic testing.

### Single-cell RNA sequencing reveals distinct features of MEPP in primary tumor and omental metastases

Given the strong metastatic potential of the MEPP model in immuno-competent mice, we sought to explore the features of TME in both primary tumors and paired omental metastases. Primary tumors and omental metastases were collected once clinical symptoms manifested and subjected to scRNA-seq (Fig. 2G). A total of 15,013 cells were analyzed, comprising 6589 cells from the primary tumors and 8424 cells from the omental metastases (Fig. 2H).

Cell clustering based on canonical markers identified a range of distinct cell populations, including epithelial cells (*Wt1*, *Mlsn*), granulosa cells (*Inha*, *Amh*, *Ihh*), macrophages (*C1qc*, *Cd68*), cDC1 (*Clec9a*, *Xcr1*), neutrophils (*S100a8*, *S100a9*), mast cells (*Tpsab1*, *Ms4a2*), T & NK cells (*Cd3g*, *Nkg7*), B cells (*Cd79a*, *Cd19*), fibroblasts (*Igftbp7*, *Rhoj*, *Cdh11*), and erythrocytes (*Rhag*, *Alas2*) (Fig. 2I, Supplementary Fig. 5B, Supplementary Data 2). Both primary and metastatic lesions were predominantly composed of epithelial cells and macrophages (Supplementary Fig. 5C). Compared to primary tumors, omental metastatic cells exhibit higher proportions of T cells and

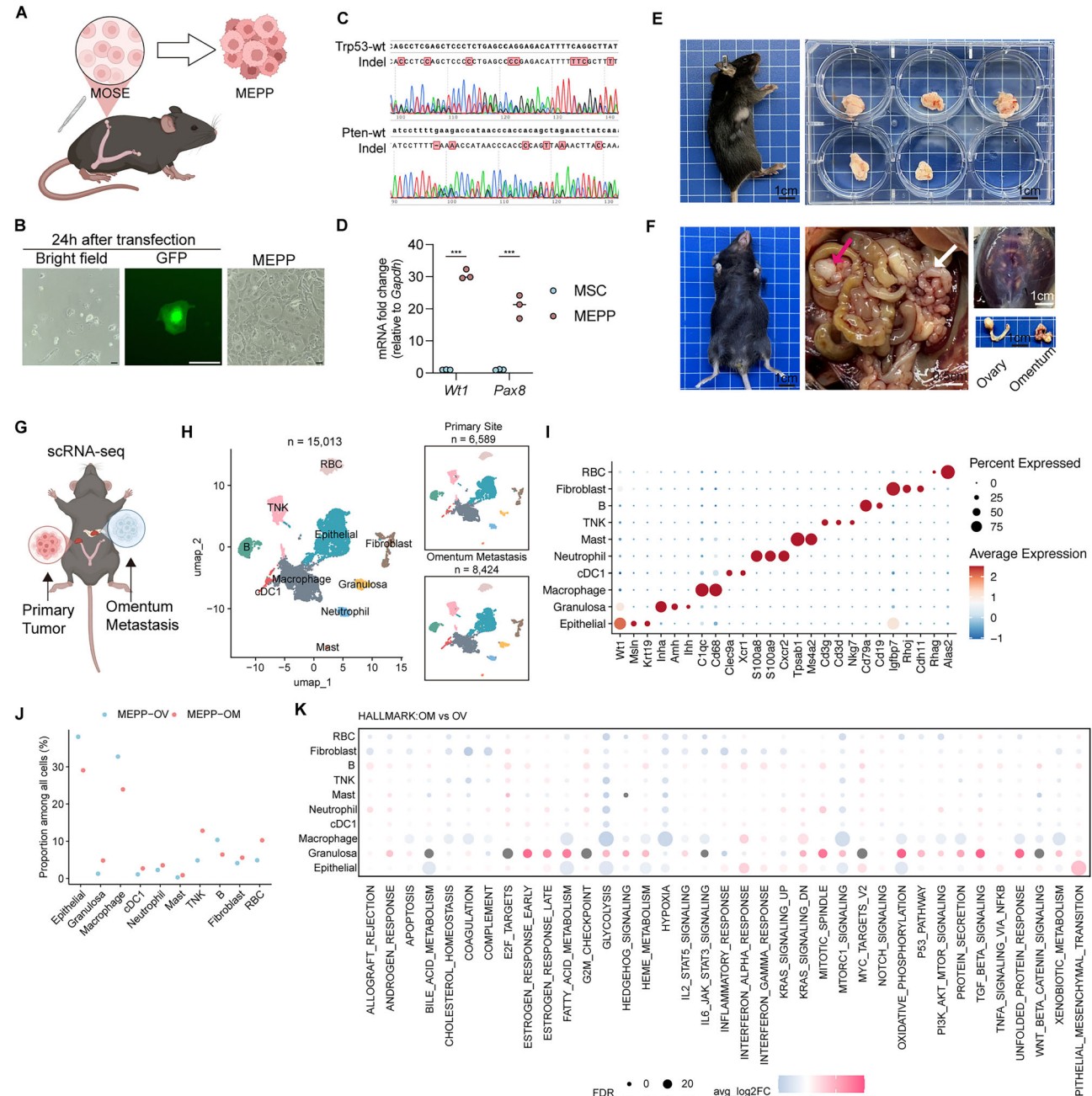

**Fig. 2 | Generation and characterization of MEPP mouse ovarian cancer model.**
**A** Schematic representation of the CRISPR/Cas9-mediated genetic engineering strategy used to knock out *Trp53* and *Pten* in MOSE. Created using BioRender.com. **B** Representative bright-field and GFP images of MOSE cells following vector transduction. Scale bar, 50 μm. **C** Sanger sequencing results confirming editing of the *Trp53* and *Pten* loci targeted by the indicated sgRNAs. **D** mRNA expression levels of *Wt1* and *Pax8* in aMSC (mouse adipose derived mesenchymal stem cell) and MEPP cells, measured by RT-qPCR and normalized to *Gapdh* mRNA expression (*n* = 3 independent experiments). **E** Representative images from the subcutaneous tumor formation experiment. **F** Representative images from the orthotopic tumor formation experiment showing tumors in the ovary (red arrow) and omentum (white arrow). **G** Schematic workflow for scRNA-seq, illustrating the collection of samples from primary ovarian (OV) and omental metastatic (OM) tumor sites. Created using BioRender.com. **H** UMAP plots depicting 6589 cells from the primary tumor and 8424 cells from the metastatic tumor, revealing a total of 10 distinct clusters. **I** Dot plots displaying the average expression of established markers in the indicated cell clusters. **J** Proportions of the 10 identified cell types across the primary and metastatic tumor sites. **K** Dot plots illustrating hallmark pathways associated with differentially expressed genes in each cell type, comparing primary and metastatic tumors.

NK cells, while showing reduced proportions of epithelial cells and macrophages (Fig. 2J).

Hallmark gene set analysis revealed several key pathway alterations between the primary and metastatic tumors (Fig. 2K). One of the most prominent changes was the enrichment of EMT in the tumor cells within the omental metastases. Additionally, immune-related pathways, such as IL-2, IL-6, and IFN signaling, were more strongly modulated in tumor cells and fibroblasts than in immune cells, indicating tumor-stromal-mediated immunological remodeling at the metastatic sites. Furthermore, metabolic pathways, including glycolysis and fatty acid metabolism, were less activated in epithelial cells and macrophages within the metastatic tumors, suggesting a shift in metabolic signatures as the tumor cells progress to metastatic sites.

Collectively, these data delineate site-specific transcriptional and compositional differences between primary and omental MEPP tumors and provide a framework for interrogating microenvironmental influences on dissemination.

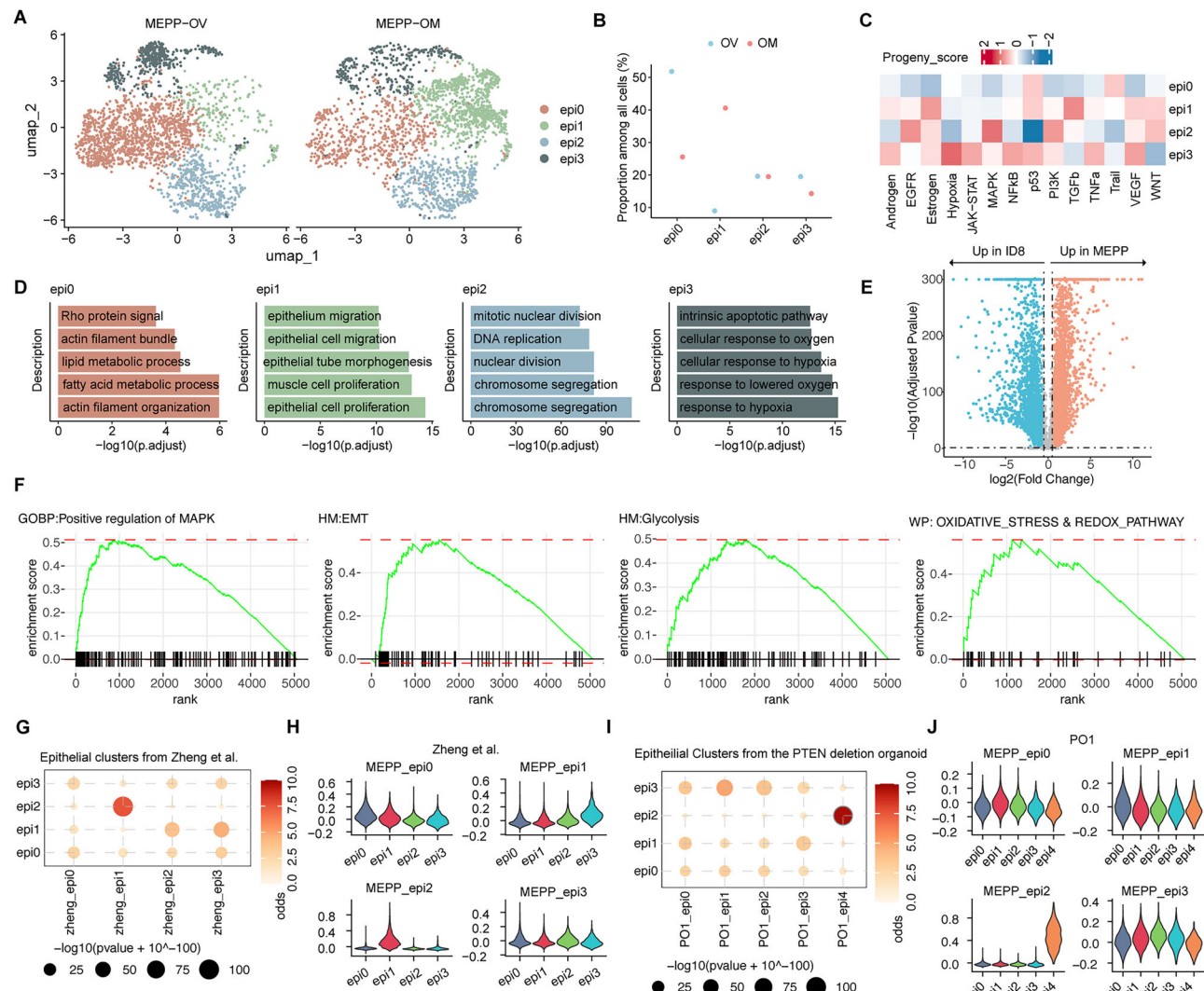

**Fig. 3 | Distinct epithelial subpopulations and malignant signatures in MEPP.**
**A** UMAP plots displaying four transcriptionally distinct tumor subclusters identified in primary ovarian (OV) and metastatic omental (OM) sites. **B** Dot plot illustrating the relative proportion of each tumor subcluster between OV and OM tumors. **C** Heatmap of mean PROGENy pathway activity scores across the four tumor subclusters. **D** Gene Ontology (GO) annotation highlighting biological processes enriched in each tumor subcluster. **E** Volcano plot of differentially expressed genes between MEPP and ID8 tumor cells. **F** Gene set enrichment analysis

(GSEA) of genes upregulated in MEPP compared with ID8, showing enrichment of hallmark pathways. **G** Cross-species comparison of MEPP tumor subclusters with published human ovarian cancer single-cell data (Zheng et al.[82]) using a hypergeometric test. **H** Violin plots showing the average expression levels of MEPP tumor subcluster–specific features in human ovarian cancer cells (Zheng et al.[82]).
**I** Comparison of MEPP tumor subclusters with the PO1 organoids using a hypergeometric test. **J** Violin plots showing the average expression levels of MEPP tumor subcluster–specific features in PO1 cells.

## Distinct epithelial subpopulations and malignant signatures in MEPP

To investigate the metastatic potential of MEPP, we utilized its capacity for metastasis to analyze the features of metastatic tumor cells. Unsupervised clustering of the epithelial tumor cells identified four distinct clusters (Fig. 3A, Supplementary Fig. 5D, E, Supplementary Data 3). Among these, the epi0 and epi1 clusters exhibited notable distribution differences between the primary tumors and omental metastases (Fig. 3B). Specifically, the epi0 cluster was predominantly enriched in primary tumors, while the epi1 cluster showed a strong preference for omental metastases.

To better understand the functional differences of these subclusters, we performed PROGENy analysis[48] and GO annotation, revealing distinct molecular signatures (Fig. 3C, D, Supplementary Data 4). The epi0 cluster demonstrated elevated apoptotic signaling and increased metabolic activity, suggesting a role in tumor survival and growth. In contrast, the epi1 cluster was associated with the activation of TGFβ and estrogen signaling pathways, along with the enrichment of migration-related pathways, indicating a

higher metastatic potential. The epi2 cluster displayed characteristics of a proliferative phenotype, while the epi3 cluster exhibited traits of hypoxia and apoptotic susceptibility, although neither cluster showed significant site-specific differences. Site-resolved indel profiling at *Trp53* and *Pten* showed no significant difference in the proportion of frameshifting mutations between the two loci (Supplementary Fig. 5F).

We compared the MEPP-OV model with ID8, a commonly used ovarian cancer model that lacks malignant mutations (Supplementary Fig. 5G). Differential gene analysis revealed significant disparities between the two models. Gene Set Enrichment Analysis (GSEA) identified pathways enriched in MEPP tumors that are related to *Pten* and *Trp53* deletions, including MAPK activation, EMT, glycolysis, and oxidative stress (Fig. 3E, F). Further analysis using hypergeometric testing and module score analysis demonstrated transcriptional concordance between MEPP clusters and human ovarian cancer tumor cell populations (Fig. 3G, H, Supplementary Fig. 5H, I). Additionally, MEPP tumor clusters exhibited transcriptomic features that closely resembled those found in *PTEN*-deletion organoids

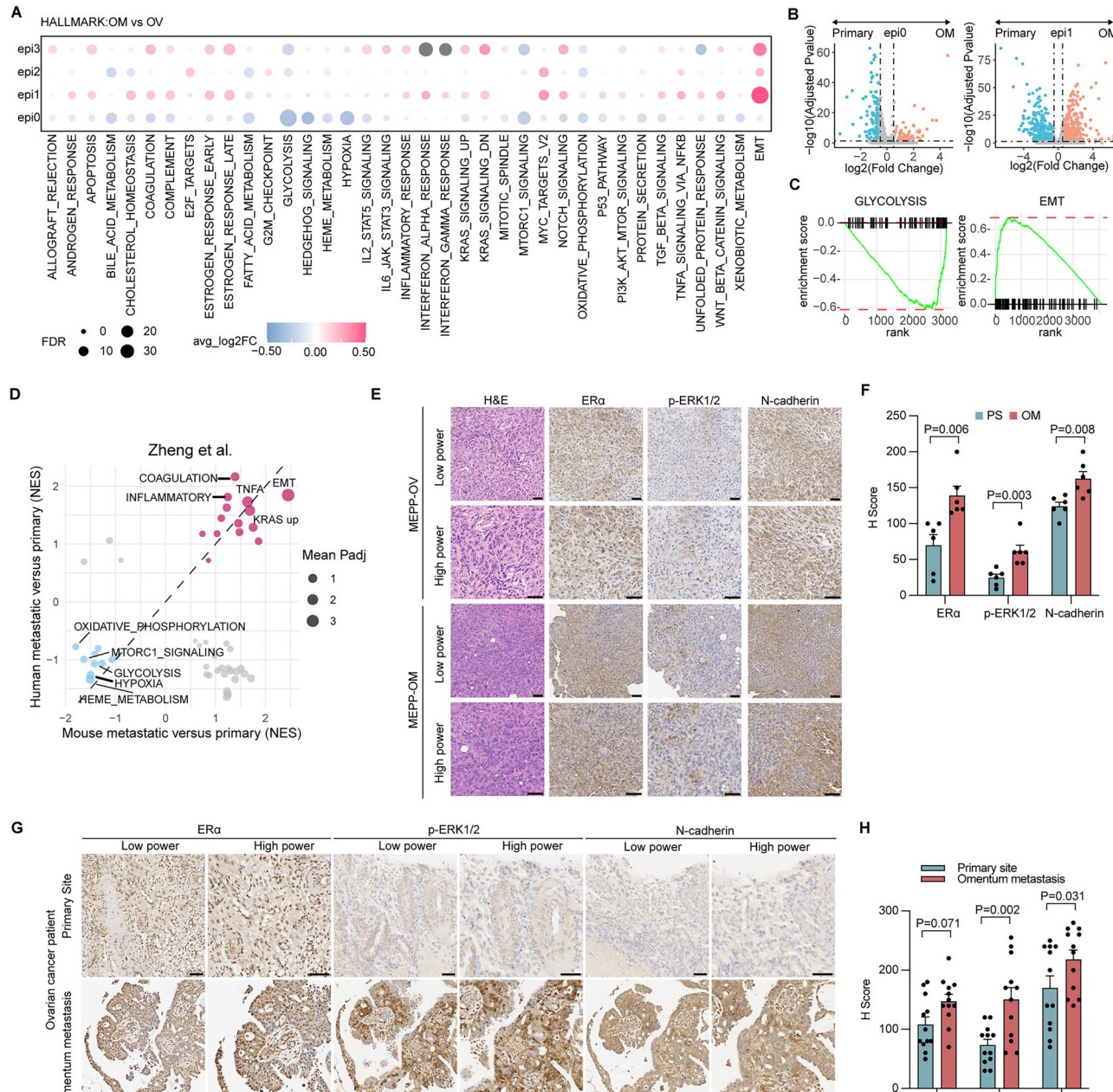

**Fig. 4 | MEPP Exhibited site-specific transcriptional difference similar to human ovarian cancers. A** Dot plots of Hallmark pathways showing differential enrichment across major cell types between primary ovarian (OV) and metastatic omental (OM) tumors. **B** Volcano plot of differentially expressed genes between tumor subclusters epi0 and epi1 at distinct anatomical sites. **C** Gene set enrichment analysis (GSEA) illustrating glycolysis enrichment in epi0 (left) and EMT enrichment in epi1 (right) in OM tumors compared with OV tumors. **D** Comparison of GSEA normalized enrichment scores (NES) for Hallmark pathways enriched in MEPP tumors and in human ovarian cancer omental metastases relative to primary tumors[82]. **E** Representative H&E and IHC staining of ERα, phospho-ERK, and N-cadherin in

MEPP OV and OM tumors. Scale bar, 50 μm. **F** Semi-quantitative IHC analysis demonstrating site-specific upregulation of ERα, phospho-ERK, and N-cadherin in OM metastases (*n* = 6 biologically independent animals) compared with OV lesions (*n* = 6 biologically independent animals). Data are mean ± s.e.m.; two-tailed Student's *t* test. **G** Representative IHC images of ERα, phospho-ERK, and N-cadherin in OV and OM tumors at both low and high magnification. Scale bar, 50 μm. **H** Semi-quantitative IHC analysis validating site-specific upregulation of ERα, phospho-ERK, and N-cadherin in human ovarian cancer OM metastases compared with OV tumors (*n* = 12 biologically independent samples). Data are mean ± s.e.m.; two-tailed Student's *t* test.

(Fig. 3I, J). This further supports the relevance of MEPP as a clinically representative model, capturing key molecular features of human ovarian cancers.

## MEPP exhibited site-specific transcriptional difference similar to human ovarian cancers

To elucidate mechanisms underlying the distinct biological features of epi0 and epi1, we performed Hallmark gene set analysis. This revealed enhanced

immune activation in omental epi1 populations, characterized by upregulation of IL2-STAT5, IL6-JAK-STAT3, and interferon signaling pathways (Fig. 4A–C). GSEA using Hallmark gene sets revealed shared transcriptional programs between MEPP and human ovarian tumors. Cross-cohort analysis showed directional concordance for activation of EMT, inflammatory response pathways, coagulation cascades, TNF-α signaling, and upregulated KRAS signaling, all of which contribute to ovarian cancer metastasis (Fig. 4D, Supplementary Data 5)[49–53].

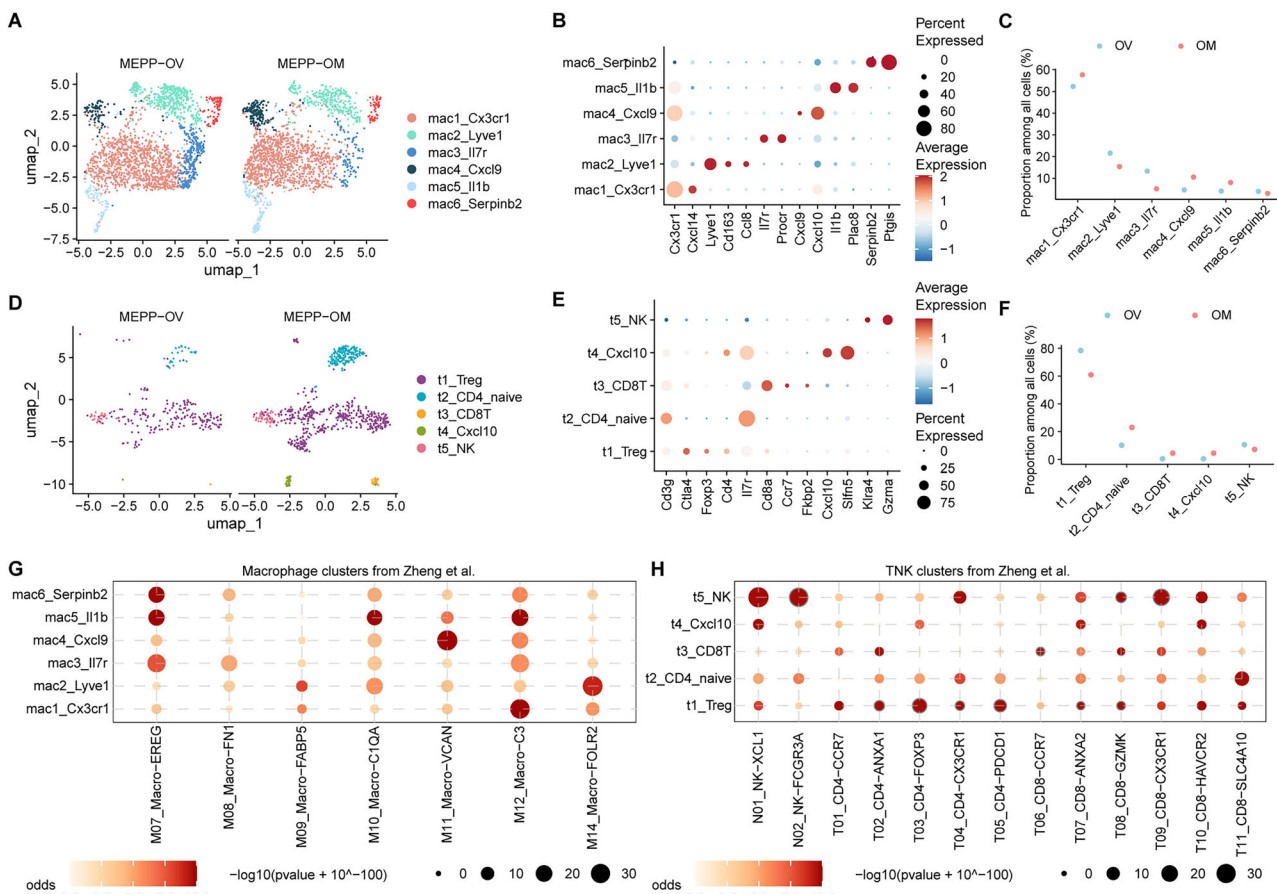

**Fig. 5 | Immune landscape characterization in MEPP tumor microenvironment.**
**A** UMAP plots identifying six macrophage subclusters in primary ovarian (OV) and omental metastatic (OM) tumors. **B** Dot plot showing the average expression of canonical marker genes across the identified macrophage subclusters. **C** Dot plot illustrating the relative proportions of macrophage subclusters between OV and OM tumors. **D** UMAP plots identifying five T/NK subclusters in OV and OM tumors.

**E** Dot plot showing the average expression of representative marker genes across the T/NK subclusters. **F** Dot plot illustrating the relative proportions of T/NK subclusters between OV and OM tumors. Cross-species comparison of tumor-infiltrating immune cells in MEPP with human ovarian cancer datasets[82] using a hypergeometric test, showing overlap of marker genes for macrophage subclusters (**G**) and T/NK subclusters (**H**).

We further conducted immunohistochemistry (IHC) analysis to characterize alterations between primary tumors and omental metastases (Fig. 4E, F). The IHC results demonstrated a marked upregulation of key oncogenic markers in the metastatic lesions. N-cadherin expression was significantly elevated, indicating enhanced EMT process. Furthermore, heightened expression of the estrogen receptor α (ERα) in these lesions suggests a potential involvement of estrogen-mediated signaling in driving metastatic progression. Notably, the metastatic lesions also displayed increased levels of phosphorylated ERK (p-ERK1/2), reflecting the hyper-activation of pivotal survival and proliferative signaling pathways. IHC of paired patient samples showed elevated N-cadherin, ERα and p-ERK1/2 in a similar pattern (Fig. 4G, H). Together, these results showed the ability of MEPP used for studying metastatic ovarian cancer.

### Conserved and divergent immune-microenvironment features in MEPP relative to human ovarian cancers

Immune compartments play a critical role in tumor metastasis. In MEPP metastatic tumors, immune cell populations were comprehensively characterized using canonical marker expression. Macrophages were stratified into six distinct transcriptomic clusters, with no significant differences in proportional distribution observed between primary tumors and omental metastases (Fig. 5A–C).

T cell populations were further classified into regulatory T cells (Tregs), naïve CD4+ T cells, CD4+Cxcl10+ T cells, and CD8+ T cells (Fig. 5D, E). Notably, Tregs were more prevalent in primary tumor sites, potentially contributing to an immunosuppressive TME that facilitates metastatic progression (Fig. 5F). Conversely, naïve CD4+ T cells showed an inverse trend, likely reflecting their differentiation into Tregs within the tumor.

Comparative cluster analysis revealed moderate similarities between the infiltrating immune cells in the MEPP model and those observed in human ovarian tumors, with each demonstrating distinct immunological characteristics across anatomical sites (Fig. 5G, H, Supplementary Fig. 6A, B). Moreover, the orthotopic model captured additional immune cell populations—such as neutrophils, B cells, and conventional dendritic cell 1 (cDC1) cells—that have emerged as crucial players in tumor metastasis (Supplementary Fig. 6C–H)[54–56]. Meanwhile, a tumor model with a complete immune microenvironment is crucial for evaluating the antitumor efficacy of immunotherapy in the future.

### MEPP served as a platform for ovarian cancer drug discovery

Given the biological fidelity of the MEPP model in vivo, we assessed its potential as a drug screening platform for ovarian cancer therapeutics. A comprehensive drug screening approach was implemented using a compound library including 328 epigenetic modulators and 1655 natural compounds (Fig. 6A). *PTEN* loss is common in prostate cancer and potently activates the PI3K–Akt–mTOR pathway, a signaling axis closely linked to tumor progression—especially in metastatic castration-resistant prostate cancer (mCRPC)[57,58]. To evaluate tissue-specific drug responses, we also generated a parallel *Trp53/Pten* knockout mouse prostate epithelial cell line (PREPP) using a methodology similar to that used for MEPP development.

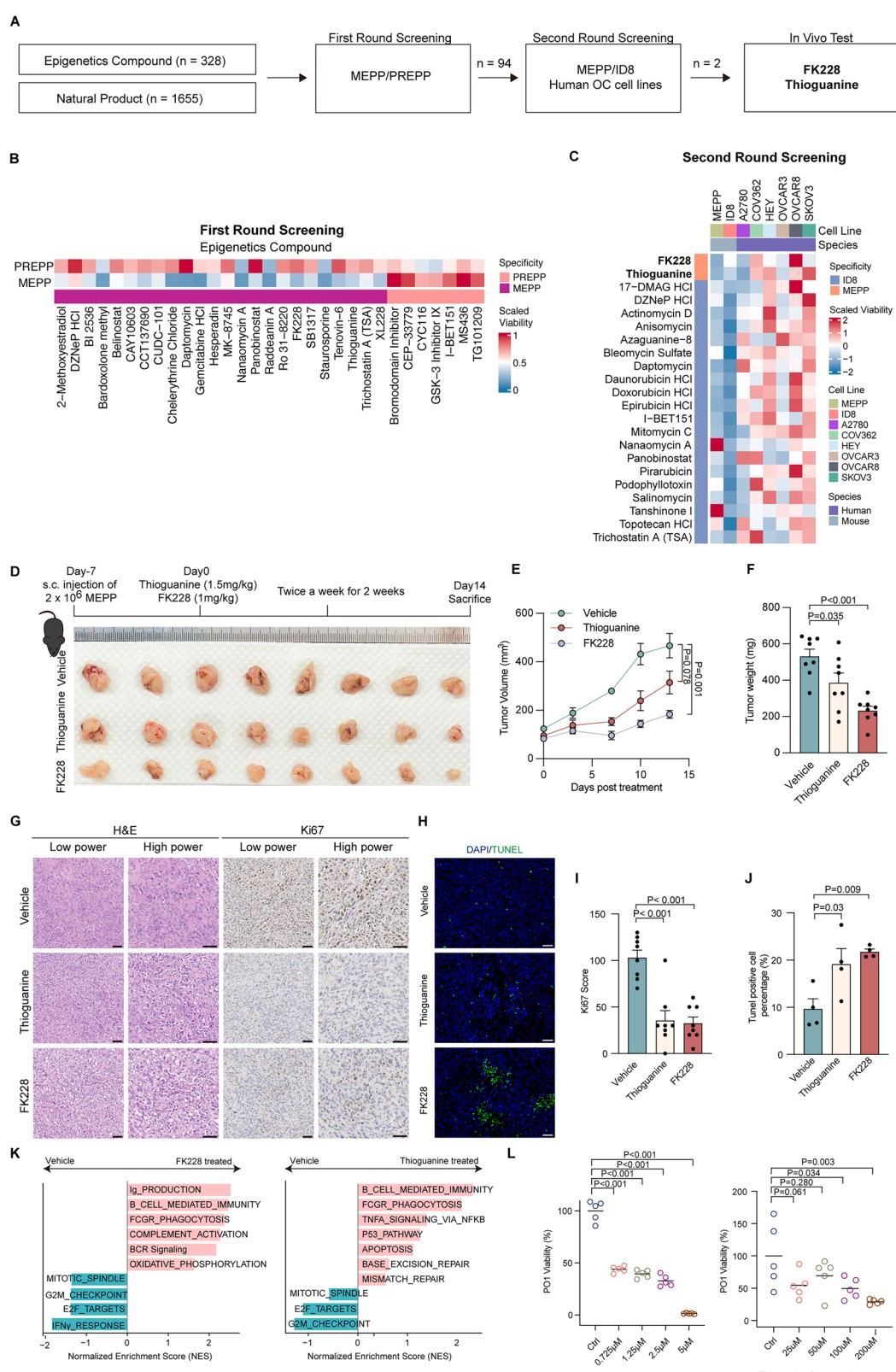

The complete drug library was screened in both MEPP and PREPP models to identify drugs that were particularly sensitive to ovarian cancer with *Trp53/Pten* deletion (Fig. 6B, Supplementary Fig. 6I, Supplementary Data 6).

The differential drug sensitivity profiles between MEPP and PREPP revealed tissue-specific therapeutic vulnerabilities. A subsequent comparative analysis between MEPP and ID8 cells, with further validation in several human ovarian cancer cell lines, identified FK228 and thioguanine as potential therapeutic candidates specific to MEPP (Fig. 6C, Supplementary Data 7). Notably, while ovarian cancer cells were resistant to doxorubicin, a clinically used chemotherapy drug, FK228 and thioguanine showed significant efficacy (Fig. 6C).

**Fig. 6 | MEPP as a drug screening platform for ovarian cancer therapeutics.**
**A** Schematic overview of the compound library screening strategy, showing the range of epigenetic and natural compounds tested. **B** Comparative drug sensitivity analysis of MEPP versus PREPP cells, identifying compounds preferentially targeting the *Trp53/Pten*-deficient background. **C** Identification and validation of FK228 and thioguanine as ovarian cancer–selective therapeutic candidates, supported by screening across multiple human ovarian cancer cell lines. **D** Experimental design for in vivo drug testing: MEPP cells were implanted subcutaneously and treated intraperitoneally with vehicle, FK228, or thioguanine (top). Representative images of tumors at necropsy are shown (bottom). **E, F** Tumor growth inhibition by FK228 and thioguanine (*n* = 8 biologically independent animals). **E** Tumor growth curves in each treatment group (mean ± s.e.m.). **F** Tumor weights at autopsy; data analyzed by one-way ANOVA with Dunnett's multiple comparisons test. **G** H&E

and Ki67 IHC staining of tumors treated with vehicle, FK228, or thioguanine. Scale bar, 50 μm. **H** TUNEL assay of tumors treated with vehicle, FK228, or thioguanine. Scale bar, 50 μm. **I, J** Quantification of Ki67 scores (*n* = 8 biologically independent samples) (**I**) and TUNEL-positive cells (*n* = 4 biologically independent biological samples) (**J**) in vehicle-, FK228-, and thioguanine-treated tumors. Data are shown in mean ± s.e.m. Statistical significance determined by one-way ANOVA with Dunnett's multiple comparisons test. **K** Gene set enrichment analysis (GSEA) of MEPP tumors treated with FK228 or thioguanine using Reactome, Gene Ontology, and Hallmark gene sets. Only pathways with FDR $q < 0.05$ are shown; representative pathways are labeled. **L** Viability of PO1 organoids following 72 h exposure to FK228 or thioguanine, analyzed by one-way ANOVA with Dunnett's multiple comparisons test (*n* = 5 independent viability experiments).

Leveraging MEPP's robust in vivo tumorigenicity, the efficacy of these compounds was further validated in vivo. Both FK228 and thioguanine reduced tumor volume in MEPP-bearing mice (Fig. 6D–F). Treated with these compounds also suppressed MEPP proliferation, as evidenced by a notable reduction in Ki67 expression and an increase in TUNEL-positive cells, indicating enhanced apoptosis (Fig. 6G–J). GSEA analysis revealed a convergent transcriptional program characterized by upregulation of immune signaling and downregulation of proliferation programs (Fig. 6K). To further confirm the therapeutic potential of FK228 and thioguanine, we validated their effectiveness using the *PTEN*-deleted organoid model, PO1. Both drugs demonstrated significant efficacy in PO1, further supporting their potential as therapeutic candidates for ovarian cancer (Fig. 6L). These results highlight the utility of MEPP as an effective and practical platform for identifying and validating therapeutic candidates, offering a valuable tool for advancing ovarian cancer treatment research.

## Discussion

Here, we present MEPP, a clinically relevant mouse ovarian cancer model generated by knocking out *Trp53* and *Pten* in primary MOSE. scRNA-seq revealed transcriptomic similarity of PO1 and ovarian cancer cells. MEPP faithfully recapitulates key features of ovarian cancer, demonstrating robust tumorigenicity and metastasis in immunocompetent mice. MEPP's versatility is highlighted by its ability to establish both subcutaneous and orthotopic tumor models. Importantly, MEPP reflects the metastatic potential and organotropism observed in human ovarian cancer. scRNA-seq analysis revealed significant similarities between the TME of MEPP and human metastatic ovarian cancers. Through comprehensive drug screening, we identified FK228 and thioguanine as promising therapeutic candidates specifically targeting both MEPP and *PTEN*-deleted PO1 organoid. These agents exhibited significant efficacy, demonstrating their potential for clinical application. Collectively, our findings confirm MEPP as a valuable model for studying the mechanisms of metastasis and for conducting targeted therapeutic investigations, particularly in the context of *PTEN*-deleted ovarian cancers.

Establishing clinically relevant tumor models that accurately capture the biological complexities of cancer is essential for advancing oncological research. Ovarian cancer, characterized by an intricate mutational landscape and significant heterogeneity, poses substantial challenges for effective modeling. Although numerous human ovarian cancer cell lines are available, their molecular fidelity to primary tumors is often limited[59]. In contrast, mouse models offer unique advantages, particularly in the investigation of tumor characteristics within an intact immune system, thereby providing more translational potential. However, conventional GEMMs have been hindered by technical complexity and lengthy development timelines[38,60,61]. Therefore, the development of in vitro models that recapitulate tissue characteristics and function as drug selection platforms is essential for progress in ovarian cancer treatment[62].

MEPP exhibits robust tumorigenicity and molecular features that closely mirror those observed in clinically relevant ovarian cancers. The importance of *PTEN* loss on ovarian cancer tumorigenesis was validated by PO1 and reinforced by MEPP with *Pten* deletion. Enrichment of MAPK,

EMT, glycolysis and oxidative stress pathways compared with wild-type ID8 cells suggested the malignant potential of MEPP. By overcoming key limitations of existing models, MEPP represents a promising preclinical tool for the study of tumor biology and the development of targeted therapeutic strategies in *PTEN* deleted ovarian cancer.

In our study, we employed an innovative approach using a compact EPI-SauriCas9 system to generate MEPP via a double knockout in primary mouse ovarian epithelial cells[46]. The use of the EPI system is essential. The EBNA1 has been demonstrated to suppress p53 in cells and show transformative effect in tumorigenesis, being associated with various cancers[63]. Recent studies have shown that EBNA1 can induce chromosomal breakage, leading to genomic instability, thereby promoting tumor initiation and progression[64]. The EPI system has been utilized for the generation of vector-free iPSC and can improve gene editing efficiency. The oncogenic potential of EBNA1 limits its broader application but may be beneficial for tumor modeling. In our experiments, transfection of primary MOSE cells with the EPI backbone carrying a scramble sgRNA did not confer long-term expansion or tumorigenicity, indicating that EBNA1 expression alone is insufficient for transformation. Likewise, disruption of both *Pten* and *Trp53* did not produce tumors in MOSE cells without the EPI backbone. By contrast, combined inactivation of *Pten* and *Trp53* using the EPI system yielded the MEPP line, which was tumorigenic in immunocompetent mice. These results indicate that the EPI vector itself is non-transforming in this context, thereby validating uses of the EPI system in cancer model generation. Further evidence is needed to validate its use in this emerging field.

Since the plasmids retain episomally and replicate during cell division, the EPI-SauriCas9 system enables the screening of highly efficient sgRNAs without the need for viral vectors, significantly shortening experimental time and serving as an ideal tool for in vitro sgRNA screening. Additionally, the gradual loss of EPI vectors facilitates the establishment of antibiotic-free tumor cell lines, thereby avoiding potential interference with clonal selection and drug selection processes in subsequent experiments[65]. Additionally, SauriCas9 is only 3.3 kb, similar to SaCas9, and shares a common sgRNA scaffold. Moreover, it recognizes a NNGG PAM, enabling a greater number of potential cleavage sites in the genome. And its' efficiency has already proved by other groups[66–68]. This makes SauriCas9 particularly suitable for screening the sgRNAs for knocking out oncogene. In the MEPP line, no editing events were detected at the predicted off-target loci, and no inter-chromosomal translocations or large deletions involving the on-target regions were observed. Overall, EPI-sauriCas9 system could have broader application prospects for tumor modeling in other organ systems.

Most ovarian cancer patients are diagnosed at advanced stages, primarily due to the cancer's metastatic nature, which is characterized by widespread peritoneal dissemination, with a particular affinity for the omentum[69]. In our study, we demonstrate the metastatic potential of the MEPP model, revealing transcriptomic features of metastatic lesions that closely resemble those observed in human metastasis. This similarity provides a robust foundation for conducting comprehensive investigations into the molecular mechanisms driving metastatic disease. Future research will employ multi-omics approaches to further elucidate the complex biology of metastasis in this model.

It is worth noting that scRNA-seq has revealed a small subset of cells with a granulosa cell origin that have also metastasized to the omentum, effectively ruling out surgical artifact as the source of these cells. Previous studies have established that activation of the PI3K pathway and loss of *PTEN* can induce transformations in ovarian granulosa cells[70,71]. Furthermore, *TP53* mutations have been implicated in the pathogenesis of granulosa cell tumors[72]. The identification of the granulosa cell component within MEPP thus provides a potentially valuable platform for investigating ovarian granulosa tumors and their unique tumorigenic features.

The current treatment paradigm for ovarian cancer predominantly relies on platinum-based chemotherapy. However, despite significant advances, many investigational therapies have failed to deliver substantial clinical benefits for ovarian cancer patients. To address this challenge, we developed two *Trp53/Pten*-knockout cell lines, MEPP and PREPP, to screen for drugs targeting tumor cell with this specific genomic background. Comparisons with the ID8 cell line, which lacked clinically relevant mutations, revealed that FK228 and thioguanine were particularly effective against MEPP and demonstrated moderate efficacy in human ovarian cancer models. Furthermore, the efficacy of the candidate drugs was validated in the *PTEN*-deleted PO1 organoid model, reinforcing the utility of MEPP as a robust drug screening platform for ovarian cancer. FK228, a histone deacetylase inhibitor (HDACi), has shown broad antitumor activity, including significant efficacy in ovarian cancer[73,74]. Similarly, thioguanine, a synthetic guanosine analog traditionally used to treat acute lymphoblastic leukemia, has recently gained attention for its selective targeting of tumors with homologous recombination defects, such as those in ovarian and breast cancers[75,76]. In MEPP, the antitumor activity of FK228 and thioguanine did not depend on reversing PTEN-driven PI3K signaling; instead, RNA-seq indicated engagement of immune/innate pathways with concurrent suppression of proliferative programs. These findings underscored the potential of MEPP as a powerful platform for ovarian cancer drug discovery and screening, supporting therapeutic development in the context of *PTEN* and *TP53* mutations.

In conclusion, the MEPP model, developed through EPI-SauriCas9-mediated *Pten* and *Trp53* deletions in MOSE, represents a clinically relevant platform for studying ovarian cancer. This model faithfully recapitulates key features of ovarian tumorigenesis, metastasis, and the TME, making it a valuable tool for understanding the molecular mechanisms driving ovarian cancer progression. The identification of promising therapeutic candidates, FK228 and thioguanine, through drug screening further demonstrates the potential of MEPP in therapeutic discovery. Moreover, the alignment of MEPP with *PTEN*-deleted organoids strengthens its clinical relevance, offering a robust system for investigating the impact of *PTEN* loss and exploring targeted treatment options. Overall, MEPP provides a comprehensive and adaptable model that holds great promise for advancing ovarian cancer research and improving therapeutic strategies.

## Methods
### Mouse lines
All animal experiments protocols were reviewed and approved by the Institutional Animal Care and Use Committee of Renji Hospital. We have complied with all relevant ethical regulations for animal use. Mouse line used in this study is C57BL/6JGpt unless otherwise stated (hereafter referred to as C57BL/6; GemPharmatech Co., Ltd, Strain NO. N000013; MGI identifier, 6314664). NOD/ShiLtJGpt-*Prkdc*[em26Cd52]*Il2rg*[em26Cd22]/Gpt (hereafter referred to as NCG; GemPharmatech Co., Ltd, Strain NO. T001475) was used for patient-derived xenograft (PDX) experiments.

### Plasmid construction
For propagation, E. coli cultures were grown in LB medium at 37 °C with constant shaking at 220–250 rpm. E. coli strain Trelief ®5α (Beijing Tsingke Biotechnology, Cat# DZC101-96B100) was used for plasmid cloning. Gibson assembly for plasmid construction was performed using the ABclonal 2× MultiF Seamless Assembly Mix (Abclonal, Cat# RK21020). All

sgRNA were inserted into plasmids vector using T4 DNA Ligase (NEB, Cat# M0202S). sgRNA target sequences are listed in Supplementary Data 8.

EPI plasmid vector is a gift from Prof. Wang Yongming (Addgene plasmid # 135960). After being incubated with HindIII, CAG promoter, SauriCas9 with NLS, T2A-Zsgreen, P2A-puro were cloned into EPI vector sequentially using Gibson assembly. Then, HU6-sgRNA scaffold was fused between KpnI and EcoRI site. SgRNA targeting each gene was inserted into the dual-BspQI site by T4 ligation. The constructed plasmid is deposited in Addgene #249830.

### Generation of MEPP and PREPP
MOSE isolation was performed using a protocol adapted from a previous study[77]. Female C57BL/6 mice from 6 to 8 weeks of age were euthanized by cervical dislocation. Ovaries were excised and washed three times with Hank's Balanced Salt Solution (BasalMedia, Cat# B410KJ). The tissues were then digested with Trpzyme (BasalMedia, Cat# S347JV) at 37 °C for 30 min. Epithelial cells were pelleted, resuspended, and cultured in a 6-well plate using Dulbecco's modified Eagle's medium (DMEM; BasalMedia, Cat# L110KJ) supplemented with 10% fetal bovine serum (FBS; ExCell, Cat# FSP500) and 1% penicillin/streptomycin (BasalMedia, Cat# S110JV) under 37 °C with 5% $CO_2$ in a humidified incubator.

Prostate epithelial cells were isolated using a method adapted from previous literature[78]. Male C57BL/6 mice from 6 to 8 weeks of age were euthanized by cervical dislocation. Mouse prostate was then dissected and washed in phosphate buffered saline (PBS; BasalMedia, Cat# B320KJ) for three times. After mincing with scissors, prostate tissue was dissociated by incubating in DMEM/F12 (Gibco, Cat# 10565018), supplemented with 5% FBS and 1:10 dilution of collagenase/hyaluronidase (STEMCELL Technologies, Cat# 07912) at 37 °C for 1 h. Dissociated tissues were spun down at $300 \times g$ for 5 min, and resuspended in 1.5 ml of TrypLE (Gibco, Cat# 12604021) for 15 min at 37 °C. After spun down at $300 \times g$ for 5 min, cells were resuspended in 1.5 ml 5 mg ml$^{-1}$ dispase (STEMCELL Technologies, Cat# 07913) supplemented with 1:10 dilution of 1 mg ml$^{-1}$ DNase I (STEMCELL Technologies, Cat# 07900). Cells were pipetted vigorously for 1–2 min. Dissociated cells were spun down at $300 \times g$ for 5 min and then cultured in a 6-well plate using RPMI 1640 (BasalMedia, Cat# L210KJ) supplemented with 5% FBS and 100U ml$^{-1}$ penicillin/streptomycin, cultured under 37 °C in a humidified incubator with 5% $CO_2$.

Once MOSE/Prostate epithelial cells reached confluency, $3 \times 10^4$ cells per well were seeded into a 12-well plate. Transfections were performed using EZ Trans (Life-iLab, Cat# AC04L092), following the manufacturer's instructions. After 24 h, the medium was replaced with culture medium, and cells were selected with 5 µg ml$^{-1}$ puromycin (YEASEN, Cat# 60209ES10) for 1 week to establish stable cell lines which were named as MEPP and PREPP, respectively.

### Cell culture
Mouse adipose mesenchymal stem cells (aMSC) were isolated from C57BL/6 mice[79]. Female C57BL/6 mice of 6–8 weeks of age were euthanized by cervical dislocation. Adipose tissues were gathered from abdominal fat pads and washed three times with PBS. Adipose tissues were then minced with scissors and incubated with 10 ml DMEM supplemented with 1:10 dilution of collagenase/hyaluronidase at 37 °C for 1 h. Cells were spun down at $300 \times g$ for 5 min. The cells were resuspended using DMEM supplemented with 10% FBS and 100U ml$^{-1}$ penicillin/streptomycin and cultured under 37 °C in a humidified incubator with 5% $CO_2$.

Cell lines used in the study included ID8 mouse ovarian surface epithelial cells (mouse; MERCK, Cat# SCC145), Hepa1-6 (mouse; ATCC, Cat# CRL-1830), A2780 (human; Cell Bank, Chinese Academy of Sciences, Cat# SCSP-5477), COV362 (human; QuiCell Biotechnology, Cat# C772), HEY (human; QuiCell Biotechnology, Cat# H1808), OVCAR3 (human; ATCC, Cat# HTB-161), OVCAR8 (human[80]), SKOV3 (human; ATCC, Cat# HTB-77) and IOSE-80 (QuiCell Biotechnology, Cat# I421). These cell lines were cultured in DMEM supplemented with 10% FBS and 100 U ml$^{-1}$ penicillin/

streptomycin at 37 °C in a humidified atmosphere with 5% $CO_2$. Cell lines were confirmed *Mycoplasma* negative and were authenticated by short tandem repeat profiling.

## Animal experiments

For subcutaneous ovarian cancer models, MEPP cells ($3 \times 10^6$) were resuspended in 50 µl PBS and mixed with Ceturegel Matrix (YEASEN, Cat # 40192ES10) (1:1 ratio). Cell suspensions were injected subcutaneously into the dorsal region of 6- to 8-week-old female C57BL/6 mice. Tumor volume was measured using calipers and calculated as 1/2 tumor length × width$^2$. The maximal tumor volume permitted by the Institutional Animal Care and Use Committee of Renji Hospital was 1500 mm$^3$. This limit was not exceeded in any of the experiments in this study. FK228 (1 mg kg$^{-1}$; MCE, Cat# HY-15149), thioguanine (1.5 mg kg$^{-1}$; MCE, Cat# HY-13765) or vehicle were administered intraperitoneally twice a week once tumors reached 50–100 mm$^3$. At the experimental endpoint, mice were euthanized by cervical dislocation, and tumor tissues were harvested for pathological analysis.

To establish an orthotopic ovarian cancer model, MEPP cells ($1 \times 10^6$) were resuspended in 10 µl PBS and mixed with Ceturegel Matrix at a 1:1 ratio. Female C57BL/6 mice were anesthetized with isoflurane and maintained under anesthesia throughout the procedure. A small incision was made at the dorsal-medial position to expose the ovary. The cell suspension was injected beneath the ovarian bursa, and the incision was sequentially closed with 6-0 sutures for both the peritoneal cavity and skin. Mice were monitored for body weight and abdominal circumference as indicators of tumor growth. A 20% increase in abdominal circumference was defined as the experimental endpoint, at which point the mice were euthanized by cervical dislocation. Primary tumors and peritoneal metastases were collected for further analysis.

## Patient samples

The study protocol involving human participants was reviewed and approved by the Ethics Committee at Renji Hospital, School of Medicine, Shanghai Jiao Tong University. Tumor bulk tissues were collected from patients diagnosed with ovarian cancer at the Department of Obstetrics and Gynecology of Renji Hospital. All specimens were obtained with patient informed consent. All ethical regulations relevant to human research participants were followed.

## Organoid culture and in vivo study

We cultured patient-derived organoid from fresh tumor samples. Briefly, tumor samples were rinsed with PBS, minced into fragments with scissors. Tumor fragments were digested in 10 ml DMEM supplemented with 1:10 dilution of collagenase/hyaluronidase and 1 mg ml$^{-1}$ dispase at 37 °C for 30 min. After centrifugation at $300 \times g$ for 5 min and supernatant removal, 5 ml TrypLE was added for further dissociation at 37 °C for 10 min. The cell suspension was passed through a 40 µm strainer and pelleted at $300 \times g$ for 5 min. 100–10,000 cells were resuspended in Ceturegel Matrix before plating into 24-well plates (40 µl per well). 400 µl One-CULTarTM ovarian cancer culture medium (OneTar, Cat# T2235-OV100) was added to each well. Organoids were cultured under 37 °C in a humidified incubator with 5% $CO_2$. The medium was changed every 3 days for up to one month.

For serial passaging experiments, organoids were passaged at a 1:3 ratio. After aspirating the culture medium, 200 µL of organoid recovery solution (YEASEN, Cat# 41421ES60) was added to dissolve and collect the Ceturegel Matrix domes. The recovered domes were incubated on ice for 15–30 min and gently pipetted to mechanically disperse organoids into small fragments. The suspension was then centrifuged at $300 \times g$ for 5 min and the pellet was resuspended in fresh Ceturegel Matrix for seeding. Organoids were cryopreserved using organoid freezing medium (YEASON, Cat# 41422ES60).

For PDX experiments, organoids were first released from Ceturegel Matrix using organoid recovery solution and then dissociated in Trypzme at

37 °C for 10–20 min with gentle pipetting to obtain near single-cell suspensions. After counting, $1 \times 10^6$ organoid-derived cells resuspended in 50 µl PBS were mixed 1:1 with Ceturegel Matrix and subcutaneously inoculated into the flanks of NCG mice.

## RNA extraction and reverse transcription-qPCR (RT-qPCR)

Total RNA of cells was extracted using RNA isolation kit (Vazyme, Cat# RC102). RNA was reverse transcribed into cDNA using EasyTranscript (Transgene, Cat# AE311-04). RT-qPCR was conducted using SYBR green method (Lablead, Cat# R0202). All results were calculated by the ΔΔct method and performed in triplicate. The primers used were as follows: *Gapdh* (Forward: CATCACTGCCACCCAGAAGACTG, Reverse: ATGC CAGTGAGCTTCCCGTTCAG), *Wt1* (Forward: GGTTTTCTCGCTCAG ACCAGCT, Reverse: ATGAGTCCTGGTGTGGGTCTTC), *Pax8* (Forward: TGCTCAGCCTGGCAATGACAAC, Reverse: ACGAAGGTGCTT TCGAGGACCA). *Trp53* (Forward: ATGGAGGAGTCACAGTCGGA, Reverse: CAGTGAGGTGATGGCAGGAT), mouse *Pten* (Forward: TGAA GACCATAACCCACCACAG, Reverse: CATTACACCAGTCCGTCCC TT), human *PTEN* (Forward: TGAGTTCCCTCAGCCGTTACCT, Reverse: GAGGTTTCCTCTGGTCCTGGTA). All primers are synthesized from Beijing Tsingke Biotechnology.

## Single-cell dissociation

scRNA-seq experiment was performed by experimental personnel in the laboratory of NovelBio Co., Ltd. The tissues were removed and kept in MACS Tissue Storage Solution (Miltenyi Biotec, Cat# 130-100-008) until processing. The tissue samples were processed as described below. Briefly, samples were first washed with PBS, minced into small pieces (~1 mm$^3$) on ice and enzymatically digested with tumor dissociation kit (Miltenyi Biotec, Cat# 130-095-929) for 40 min at 37 °C, with agitation. After digestion, samples were sieved through a 70 µm and 40 µm cell strainer, and centrifuged at $400 \times g$ for 5 min. After the supernatant was removed, the pelleted cells were suspended in red blood cell lysis buffer (Miltenyi Biotec, Cat# 130-094-183) to lyse red blood cells. After washing, the cell pellets were resuspended and then stained with AO/PI for viability assessment using Countstar Fluorescence Cell Analyzer. The single-cell suspension was further enriched with a MACS dead cell removal kit (Miltenyi Biotec, Cat# 130-090-101).

## Single-cell RNA sequencing

The NovelCyto Single-Cell Analysis System (NovelBio Co., Ltd) was employed to capture transcriptome data by first distributing a single-cell suspension randomly across over 100,000 microwells through a limited dilution method, followed by the addition of oligonucleotide barcoded beads until saturation was achieved, ensuring that each bead paired with a cell in a microwell. After cells were lyzed in the microwells, mRNA molecules hybridized with the barcoded capture oligos on the beads. These beads were then gathered into a single tube for reverse transcription and ExoI digestion. During cDNA synthesis, each cDNA molecule was tagged at the 5′ end (corresponding to the 3′ end of the mRNA transcript) with a unique molecular identifier (UMI) and a cell barcode, which was used as cell origin. The whole transcriptome library was constructed with the NovelCyto single-cell whole-transcriptome amplification (WTA) workflow, including random priming and extension, amplification PCR, and WTA library index PCR. The libraries were quantified with a High Sensitivity DNA chip (Agilent) on a Bioanalyzer 4200 and a Qubit High Sensitivity DNA assay (Thermo Fisher Scientific). All libraries were sequenced by DNBSEQ-T7 Sequencer (MGI) on a 150 bp paired-end run.

## Single-cell RNA data processing

scRNA-seq data analysis was performed by NovelBio Co.,Ltd. with NovelBrain Cloud Analysis Platform. We applied fastp (v0.21.0) with default parameter filtering the adaptor sequence and removed the low-quality reads to achieve the clean data. To quantify the gene expression of the

single-cell data, we used STARsolo (v2.7.10a) along with mouse genome mm10 (Ensemble annotation version 100). The filtered matrix was used to detect doublets using scrublet (v0.2.3) with default parameters[81].

## Identification of cell populations

Major cell type assignments were assigned according to reported cell markers: epithelial cells (*Wt1, Mlsn*), granulosa cells (*Inha, Amh,* Ihh*)*, macrophages (*C1qc, Cd68*), cDC1 (*Clec9a, Xcr1*), neutrophils (*S100a8, S100a9*), mast cells (*Tpsab1, Ms4a2*), T & NK cells (*Cd3g, Nkg7*), B cells (*Cd79a, Cd19*), fibroblasts (*Igftbp7, Rhoj, Cdh11*), and erythrocytes (*Rhag, Alas2*). Cell subtype identifications were based on graph-based unsupervised clustering. Briefly, Louvain algorithm was performed at different resolutions (0.1, 0.2, 0.3, 0.4 and 0.5) in each superset. Differential gene expression was implemented by Seurat FindMarkers. Cluster annotations were based on marker genes identified in the gene expression analysis.

## Differential expression and functional annotation

To identify significantly overexpressed marker genes for clusters, the FindAllMarkers function of Seurat was utilized. Genes with an adjusted *P*-value < 0.05, determined by the Wilcoxon rank-sum test, were designated as cluster-specific signature genes. Genes with adjusted *P*-values < 0.05 were deemed differentially expressed and subsequently utilized for GO enrichment analysis with the clusterProfiler package (v4.2.2). GO terms with adjusted *P*-values < 0.05, using the BH procedure, were considered statistically significant. We applied GSVA for scoring analysis and functional gene sets in tumor cells.

## Cluster similarity between MEPP and human ovarian cancer subclusters

The similarity between ID8 cells and human ovarian cancer subclusters were identified using a similar method as in Vázquez-García et al.[41]. Gene lists from human ovarian cancers were obtained from publicly available gene sets from Zheng et al.[82] and Chai et al.[40]. Human gene sets were first converted to mice genes by homologene (v1.4.68.19.3.27). For comparison with tumor clusters in Zheng et al.[82], MEPP tumor feature genes list was obtained by FindAllMarkers function. Then, the two gene sets were compared for independency by using GeneOverlap (v1.30.0). Fisher's exact test were employed to identify significant related gene sets. An odds ratio more than 1 suggested a strong association between two clusters. AddModuleScore function in Seurat was applied to calculate the average expression value of each gene signature in our data. An expression value more than 0 suggested a correlation between two clusters.

## Gene set enrichment analysis

GSEA analysis[83] was performed using fgsea[84] for single-cell sequencing data using default parameters. Hallmark gene sets were available on MSigDB database (https://software.broadinstitute.org/gsea/msigdb).

## Tissue distribution of clusters

Tissue distribution of cell clusters were estimated by the ratio of observed to expected cell numbers ($R_{o/e}$) as previously reported[85,86].

## Whole genome sequencing

Genomic DNA was isolated from tissues using the Universal Genomic DNA Extraction Kit (Magnetic Bead Method; Maggi Yuhua, China) following the manufacturer's protocol. DNA was sheared to ~400 bp (Covaris) and subjected to end repair, A-tailing, and adapter ligation. Adapter-ligated fragments were PCR-enriched and then hybridized in solution with biotinylated capture probes. Targeted fragments were recovered using streptavidin magnetic beads and further amplified. Final libraries were purified (AMPure XP; Beckman Coulter) and quality-controlled (Agilent Bioanalyzer 2100). Indexed libraries were pooled and sequenced on an Illumina NovaSeq 6000 (paired-end 150 bp) according to the manufacturer's instructions. Raw reads were filtered with fastp to remove adapters and low-quality reads (mean Phred <Q20) and reads with >10% ambiguous bases.

Clean reads were aligned to hg38 using BWA-MEME software[87] with default parameters. Following the modified GATK Best Practices[88], aligned BAMs were sorted with samtools[89] and PCR duplicates were marked by MarkDuplicates. Germline variants (SNPs/indels) across samples were called using Haplotyper and Gvcftyper program in sentieon genomics tools. Somatic variants were called using Mutect2 module in sentieon genomics tools[90]. The structural variants identified in WGS are listed in Supplementary Data 9.

## Drug screening using compound library

Two drug libraries (APExBIO, Cat: L1029 and L1039P) were utilized to systematically screen for epigenetic and natural compounds with sensitivity to MEPP. Briefly, MEPP and PREPP cells were seeded at 5000 cells per well in 96-well plates. Compounds were added at a final concentration of 5 μM per well, and cells were cultured for 3 days. Cell viability was assessed using the CCK8 assay (Life-iLab, Cat# AC11L054). Drug specificity was defined as a reduction in cell viability by at least 1.5-fold compared to the control cell line and by 0.75-fold relative to the untreated group.

For the drug viability test using PO1 organoids, ~5000 organoid-derived cells were seeded per well in a 96-well plate, and drugs were added at the indicated concentrations. Cell viability was measured 24 h later using a CCK-3D kit (APExBIO, Cat# K2270) according to the manufacturer's instructions.

## Immunochemistry staining and TUNEL assay

Tissue sections were deparaffinized and rehydrated. Antigen retrieval was performed by incubating the sections in Tris-EDTA antigen retrieval buffer (Proteintech, Cat# PR30002) using a microwave oven. After cooling to room temperature, sections were blocked using blocking solution (CST, Cat# 15019) and incubated overnight at 4 °C with the following primary antibodies: WT1 (1:200; abcam, Cat# ab89901), Ki67 (1:200; abcam, Cat# ab16667), N-cadherin (1:200; Proteintech, Cat# 22018-1-AP), p-ERK1/2 (1:100; CST, Cat# 9101), and ERα (1:100; abcam, Cat# ab32063). HRP-conjugated secondary antibody (1:200; abcam, Cat# ab6721) was applied for 1 h at room temperature. DAB (YEASON, Cat# 36302ES01) was used for immunodetection. Sections were counterstained with hematoxylin, dehydrated and mounted. IHC semi-quantification was assessed by scoring staining intensity (0 = negative, 1 = weak, 2 = moderate, 3 = strong) and measuring the percentage of positive cells (0–100%). The final score was calculated by multiplying intensity by the percentage of positive cells.

For TUNEL assay, the slides were deparaffinized and rehydrated as above and processed using the TUNEL detection kit (YEASEN, Cat# 40306ES60) according to the manufacturer's instructions. TUNEL-positive cells were calculated using ImageJ2 (v2.14.0).

## Statistics and reproducibility

Statistical analyses were performed using R (v4.1.0), python (v3.6.1) and Graphpad Prism (v10.0). Statistical tests used in this study included two-sided Student's *t* tests, Wilcoxon rank-sum tests, one-way ANOVA and two-way ANOVA, as appropriate, and are specified in the corresponding figure legends. *P* < 0.05 was considered statistically significant. The number of biological replicates for each experiment is reported in the figure legends. Single-cell RNA-seq analysis were processed using established workflows and quality control criteria as described in the Methods.

## Reporting summary

Further information on research design is available in the Nature Portfolio Reporting Summary linked to this article.

## Data availability

The raw sequence data for mice reported in this paper have been deposited in the Genome Sequence Archive[91] in National Genomics Data Center[92], China National Center for Bioinformation / Beijing Institute of Genomics, Chinese Academy of Sciences (GSA: CRA024029; CRA029112; CRA034026) that are publicly accessible at https://ngdc.cncb.ac.cn/gsa. Raw

sequencing data of the human organoid have been deposited under accession HRA013088 in National Genomics Data Center (https://ngdc.cncb.ac.cn/search) and can be applied via the portal following procedure. For public datasets used in this study, scRNA-seq data from Zheng et al. are available in Mendeley Data (https://doi.org/10.17632/rc47y6m9mp.1)[82,93]. scRNA-seq data of wild type ID8 are available at GSE183368[94]. Processed matrix, source data of the figures and a table documenting method sources are available on figshare (https://doi.org/10.6084/m9.figshare.30741540). All other data are available from the corresponding author on reasonable request.

## Code availability
Analysis codes required to reproduce the results are available on figshare (https://doi.org/10.6084/m9.figshare.30741540).

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

## Acknowledgements

We appreciate the support of the CW Chua Lab at Renji Hospital for their assistance and support in this research. W.T.C. would like to thank Yanhua Du at Shanghai Jiao Tong University School of Medicine for her mentorship in scRNA-seq analysis. We appreciate Prof. Wang Yongming for the kind gift of the EPI system and SauriCas9 vector. This work was supported by the National Natural Science Foundation of China (Nos. 82172918, 82201491, 82403651, 82471723), the Shanghai Municipal Health Commission (202340083), the Medical-Engineering Joint Funds of Shanghai Jiao Tong University (YG2021GD01, YG2025ZD19), the Clinical Scientific Research Innovation Cultivation Fund of Renji Hospital (PYI20-03), the Shanghai Public Health Excellent Discipline Leadership Program (GWVI-11.2-XD15), the Renji Hospital Science and Technology Achievement Transformation Incubation Project (RJZH25-009), the Noncommunicable Chronic Diseases-National Science and Technology Major Project (2024ZD0530601) and the Science and Technology Commission of Shanghai Municipality (23JC1403000).

## Author contributions

Conceptualization: W.T.C., P.J.H., Y.W., W.D., X.P.K., and B.Y.; Methodology: P.J.H., L.D., W.H.L., Z.Z.Y.F.,Y.M.S, Y.W., and B.Y.; Investigation: W.T.C., P.J.H., L.D., W.H.L., W.W.S., Z.Z.Y.F., J.L., Z.X.T., and Y.S.W.; Validation: Y.W., W.D., B.Y., and X.P.K.; Visualization: L.D., Y.S.W., and Z.Z.Y.F.; Writing – Original Draft: W.T.C., P.J.H., L.D., W.H.L., and Y.S.W.; Writing – Review & Editing: Y.W., W.D., X.P.K., and B.Y.; Supervision: Y.W., W.D., X.P.K., and B.Y.

## Competing interests

The authors declare no competing interests.
