## [Transparent Peer Review file · Communications Biology]

EPI-SauriCas9-Based Mouse Ovarian Cancer Models Recapitulating PTEN Deletion in Patients

Corresponding Author: Professor Bin Yu

Version 0:

Reviewer comments:

Reviewer #1

(Remarks to the Author)

The manuscript from Chen W. et al. describes the generation of the MEPP cell model by double knockout of Pten and Trp53 to recapitulate frequent genetic alterations found in ovarian cancers. The cell model was established with EPI-SauriCas9-dual-sgRNA system for episomal replication of the plasmid and the long-term expression of the nuclease. The authors performed in depth analysis of tumorigenicity and metastatic potential, showing the potential utility of MEPP cells in replicating the ovarian cancer features in term of tumor growth, immunogenicity and metastasis in the omentum.

The authors performed scRNAseq to explore the cell population and expression of markers in the primary tumors and in omental metastases, showing shared features and differences between the two lesion types, and demonstrating that MEPP cells recapitulate the cellular and molecular key aspects of ovarian cancer. Finally, the authors used MEPP as a drug screening platform and showed that FK228 and thioguanine are effective in targeting MEPP cells, with modest effect on human ovarian cancer cells and human organoids.

The results presented by the authors show the potential application of MEPP as a model for ovarian cancer in mechanistic studies and drug screening. However, further evidence should be provided to validate the utility of MEPP as an effective model for PTEN-derived ovarian cancers.

Major revisions:

1) Editing efficiency of Pten and Trp53 loci in Hepa1-6 cells should be shown in terms of percentage of indels.

2) A detailed characterization at the genomic level of MEPP cells should be provided:

- frequency of indels at the targeted loci;
- percentage of cells that are KO for both loci;
- percentage of frameshift indels;
- downregulation of expression of target loci should be analysed;
- Off-target analysis, including chromosomal rearrangements that could occur when multiplex editing is performed;
- As negative control, transfection of MOSE cells with EPI-SauriCas9 system coupled to scramble sgRNA should be performed to consider EBNA1 transformative effect in MEPP.

These data will be instrumental to demonstrate that MEPP cells are homogeneous for KO of Pten and Trp53. Moreover, these data will show that the phenotype as well as the tumorigenic and metastatic potential of MEPP cells are related only to double knockout of Pten and Trp53, and not to targeting of off-target sites or EBNA1 expression.

3) In case of heterogeneity of MEPP cells, it could be speculated that the development of tumor subclusters identified in primary or metastatic sites is driven by loss of Pten and/or Trp53. The analysis of indels in primary and metastatic sites would be highly informative for a precise characterization of MEPP cells as ovarian cancer model.

4) Semi-quantitation of IHC of staining of MEPP (Fig.5E) should be performed to confirm the comparison with IHC staining in primary and omental metastatic tissues.

5) The use of MEPP as a drug screening platform does not seem to fully recapitulate the sensitivity of human ovarian cancer cells to therapeutic candidates. Statistical analysis of PO1 drug sensitivity is required to support the use of MEPP for drug screening. Quantification of TUNEL assay of tumors should also be performed.

Minor comments:

- 1) The sequence of sgRNA used in the study should be provided.
- 2) Figure legends should be explained with further details
- 3) Rephrase sentences, lines 126-128 and 257-259.

Reviewer #2

(Remarks to the Author)

Authors: Wutao Chen, Pengju He, Ling Ding, Weihua Lou, Yishu Wang, Weiwei Shi, Zhangzhengyi Fan, Yumeng Sheng, Jing Luo, Zhixing Tan, You Wang, Wen Di, Xiaoping Ke, Bin Yu

Title: EPI-SauriCas9-Based Murine Ovarian Cancer Models Recapitulating PTEN Deletion in Patients

Manuscript ID: COMMSBIO-25-3222-T

The manuscript presents MEPP (Murine Ovarian Epithelium with Pten and Trp53 deletions), a novel murine ovarian cancer cell line developed using the innovative EPI-SauriCas9 gene-editing platform. MEPP is positioned as an immunocompetent, genetically stable model that mirrors the molecular landscape of human ovarian cancer. This model is suggested as superior to traditional ovarian cancer organoids due to its ability to sustain long-term cultures, generate metastasizing tumours in immunocompetent mice and therefore incorporate immune and stromal interactions - key limitations of current organoid models. Importantly, MEPP is proposed as a robust platform for high-throughput drug discovery, validated through successful *in vivo* efficacy studies with FK228 and thioguanine. It is noteworthy that, although MEPP can be used as a cell line, its full benefit (such as immunocompetence) only emerges in the *in vivo* situation, which makes it a model that mimics human ovarian cancer but does not replace animal research. While the manuscript successfully highlights the potential of MEPP for the field of ovarian cancer biology, issues in logical flow and some overextensions of conclusions limit the quality and impact of the study.

Major points:

- 1) The manuscript opens with a characterization of patient-derived ovarian cancer organoids, highlighting PTEN exon 8 mutations sustaining long-term growth of one specific organoid line, PO1. As the Introduction is constructed to highlight the previously reported limitations of organoid models and the necessity to develop a better model system, it is surprising to find Fig.1 focussed merely on organoids rather than directly starting with the development of the MEPP model. The rationale for targeting Pten and Trp53 stems from their critical roles in ovarian cancer pathogenesis and their role in previous model systems, as highlighted extensively in the Introduction. In that sense, the data presented in Fig.1 appear as Supplementary Data to the actual focus of the manuscript, which is the establishment and characterization of the MEPP model, and better streamlining of the manuscript should be considered to strengthen positive aspects of the MEPP model and remove repetitive elements in the text. At the very least, an introductory sentence clarifying the purpose of studying organoids initially would strengthen the logical flow.
- 2) Lines 95/96 state "long-term propagation achieved for only a few organoid lines (Fig.1A)". Were there organoid lines other than PO1 that sustained long-term growth? If so, why have they not been included in the analyses in Fig.1? Are they all PTEN-deficient?
- 3) The authors characterize the mutational profile of PO1 by Sanger sequencing to find PTEN exon 8 mutations and state that these result in deletion of PTEN. No expression data are shown to support this claim.
- 4) The authors then present EPI-SauriCas9 as a novel dual-sgRNA plasmid for efficient knockout of Pten and Trp53. While innovative, the novelty of this plasmid system is not sufficiently differentiated from existing episomal and SauriCas9 systems (lines 126-133). Moreover, the data presented in Fig.2 appear supplementary to MEPP development, utilizing a murine hepatocarcinoma line (Hepa1-6) rather than ovarian models. It should be considered to move Fig.2 to the Supplementary Information.
- 5) Fig. 3 marks the true beginning of the manuscript's focus: the development and *in vivo* characterization of the MEPP model. The data convincingly demonstrate MEPP's ability to metastasize to the omentum in an orthotopic mouse model. scRNA-seq of primary tumours and metastatic sites provides insight into the cellular composition of the primary tumour, metastasis and tumour microenvironment. The data are largely descriptive and the section titles seem to promise more than the data support. They should be rephrased to what is really demonstrated, e.g. distinct features of primary tumour and metastases in MEPP rather than distinct metastatic features (line 169).
- 6) Lines 210-212 state: "Gene Set Enrichment Analysis (GSEA) identified pathways enriched in MEPP tumors that are related to PTEN and Trp53 deletions, including MAPK activation, EMT, glycolysis, and oxidative stress (Fig.4E-F)". MEPP is a different model from ID8, as I understand it. Are they derived from the same starting cells (MOSE)? Otherwise, the difference in the pathways enriched in MEPP compared to ID8 might not/not only be attributed to the presence of Pten and p53 deletions.

7) Upregulation of N-cadherin at metastatic sites as shown in Fig. 5F is expected and not indicative of the MEPP model mimicking human ovarian cancer metastatic features, as the section title suggests. Even if found in both MEPP metastases and human ovarian cancer metastases, it represents a general marker of EMT/metastasis and does not justify the conclusion that the MEPP model specifically mimics ovarian cancer metastasis. The only data that would justify this conclusion is the data presented in Fig. 5D, but there are also pathways that are not aligned – is the conclusion of a linear relationship justified? Moreover, looking at Suppl. Fig.S4, not all primary tumor/metastasis pairs show the increased N-cadherin, ER and p-Erk in the omentum metastatic sites. Are the stainings shown in Fig. 4 representative?

8) Lines 247-249 state “Comparative cluster analysis revealed moderate similarities between the infiltrating immune cells in the MEPP model and those observed in human ovarian tumors (Fig. 6G-H, Suppl. Fig. 5A-B)”. As the similarities are moderate only, the title of the section “MEPP recapitulates the human tumor microenvironment” appears overstated in relation to the presented data and should be rephrased. The section would also benefit from enhanced cross-comparisons with human datasets and experimental validation.

9) MEPP's utility in therapeutic screening is convincingly validated by the identification of FK228 and thioguanine, which demonstrate significant efficacy in vivo. It would be interesting to include how these drugs work. Do they act on the PTEN deficiency/PI3K activation?

Minor points:

- It is unclear what is IOSE in Fig. 1D, not mentioned in the text. An ovarian cancer cell line?
- Fig. 1D, 2D: wildtype sequence missing to compare to
- Line 120: providing should be provided
- line 126: knocking out should be knockout
- Line 138: adding should be addition
- The significance of Suppl. Fig.S2B is unclear and the figure is not sufficiently described.
- Remove introductory paragraph to Fig.4 (lines 194-196), as Fig.3 already analysed metastasis in the MEPP model, no need to introduce it again.
- Line 251: Fig.S4C-H should be Fig. S5C-H

Version 1:

Reviewer comments:

Reviewer #1

(Remarks to the Author)

The authors have provided convincing and detailed evidence about genome editing profile of MEPP cells, as well as primary tumors and omental metastasis. Moreover, they have further supported the use of MEPP for drug screening. I appreciate the effort invested in addressing my comments, and I am satisfied that all my concerns have been fully resolved.

Reviewer #2

(Remarks to the Author)

Authors: Wutao Chen, Pengju He, Ling Ding, Weihua Lou, Yishu Wang, Weiwei Shi, Zhangzhengyi Fan, Yumeng Sheng, Jing Luo, Zhixing Tan, You Wang, Wen Di, Xiaoping Ke, Bin Yu

Title: EPI-SauriCas9-Based Murine Ovarian Cancer Models Recapitulating PTEN Deletion in Patients

Manuscript ID: COMMSBIO-25-3222-T

The authors have made a commendable effort to answer my earlier concerns, and I believe the manuscript has been improved significantly. However, a few remaining issues should be addressed before the manuscript is suitable for publication:

Supplementary Table S1: The table does not indicate the PTEN status of the cell lines, as stated in the rebuttal letter.

Including this information would be valuable for the reader and would support the manuscript's emphasis on the importance of PTEN.

Figure 6L: Please clarify how viability was assessed. This should be described in the text or figure legend, or included in the methods section. For the PO1 viability curve with FK228, the inclusion of lower doses would strengthen the data. The presented dose of 5 μ M already results in maximum efficacy and complete loss of cell viability. Testing and presenting lower doses would help determine whether partial viability is maintained at submaximal concentrations.

Response Letter

Manuscript title: EPI-SauriCas9-Based Murine Ovarian Cancer Models Recapitulating PTEN Deletion in Patients

We would like to thank all reviewers for your time and valuable comments. In this major revision, we have added necessary experiments and analysis as required to address points asked. All changes made to the manuscript are marked in red.

Review#1

Q1: Editing efficiency of *Pten* and *Trp53* loci in Hepa1-6 cells should be shown in terms of percentage of indels.

R1: Thank you for this constructive comment. We have now quantified and presented the indel frequencies of *Pten* and *Trp53* loci in Hepa1-6 cells. Both sgRNA pair1 and sgRNA pair2 showed comparable indel frequencies, whereas sgRNA pair3 did not yield detectable indels. We further assessed the editing efficiencies of sgRNA pair1 and sgRNA pair2 in the murine ovarian cancer cell line ID8, and found that sgRNA pair1 exhibited the highest editing frequency. Based on these results, we selected sgRNA pair1 for all subsequent experiments (Response Fig.1).

Response Fig.1 Editing frequency of different sgRNA pairs targeting *Trp53* or *Pten* in Hepa1-6 (A) or ID8 (B).

Q2: A detailed characterization at the genomic level of MEPP cells should be provided:

- frequency of indels at the targeted loci;
- percentage of cells that are KO for both loci;
- percentage of frameshift indels;
- downregulation of expression of target loci should be analysed;
- Off-target analysis, including chromosomal rearrangements that could occur when multiplex editing is performed;
- As negative control, transfection of MOSE cells with EPI-SauriCas9 system coupled to scramble sgRNA should be performed to consider EBNA1 transformative effect in MEPP.

These data will be instrumental to demonstrate that MEPP cells are homogeneous for KO of *Pten* and *Trp53*. Moreover, these data will show that the phenotype as well as the tumorigenic and metastatic potential of MEPP cells are related only to double knockout of *Pten* and *Trp53*, and not to targeting of off-target sites or EBNA1 expression.

R2: Thanks for your comments. We have performed experiments to answer the above questions.

(1) Frequency of indels at the targeted loci.

Amplicon deep sequencing of early-passage MEPP single-cell clones (n = 10) demonstrated high on-target editing efficiency. Specifically, 80% of clones exhibited biallelic deletions in *Trp53*, and all clones exhibited biallelic deletions in *Pten* (Response Fig.2A). Two clones showed monoallelic deletions, which we attribute to clonal variability. Because sequencing technical errors can result in indel frequencies clustering around ~50% (monoallelic) or ~100% (biallelic), the observed values are consistent with the expected allelic editing patterns. Together, these data confirm that the editing efficiency was high in early-passage MEPP cells.

(2) Percentage of cells KO at both loci.

We isolated 40 early-passage MEPP single-cell clones, PCR-amplified the *Pten* and *Trp53* target sites, and Sanger-sequenced the amplicons. All 40/40 clones carried frameshifting indels in both genes, indicating 100% double-knockout among the sampled clones (Response Fig.2B).

(3) Percentage of frameshift indels.

From the amplicon sequencing of 10 early-passage clones, nearly all indel events resulted in frameshifts. Frameshift events accounted for ~100% of the detected indels, and the percentage of frameshift events among total reads was therefore comparable to the overall indel frequency (Response Fig.2C).

(4) Down regulation of target loci.

We have now quantified transcript levels of the two deleted tumor-suppressor loci and their canonical downstream programs. qPCR analysis using primers spanning targeted loci was used to confirm depletion of each transcript in MEPP versus MOSE (Response Fig.2D).

Trp53 Primer: F-5'- ATGGAGGAGTCACAGTCGGA -3'; R-5'- CAGTGAGGTGATGGCAGGAT - 3'

Pten Primer: F-5'-TGAAGACCATAACCCACCACAG-3'; R-5'- CATTACACCAGTCCGTCCCTT -3'

GSEA on the ranked transcriptome (MEPP vs MOSE) demonstrates (i) strong positive enrichment of HALLMARK_MTORC1_SIGNALING and HALLMARK_MYC_TARGETS_V1, consistent with PI3K-AKT-mTOR activation following PTEN loss; and (ii) coordinated up-regulation of REACTOME_CELL_CYCLE_CHECKPOINTS, HALLMARK_E2F_TARGETS and HALLMARK_G2M_CHECKPOINT, reflecting p53-checkpoint failure (Response Fig.2E).

(5) Off-target analysis.

We first queried the CRISPR-Cas9 off-target server CCTop (<https://cctop.cos.uni-heidelberg.de/>) and retrieved murine genomic sites harboring ≤ 3 mismatches to either gRNA (sgPten or sgTrp53). The seven targets (*Cnot1*, *Fndc3b*, *9230020A06Rik*, *Nedd4l*, *Ano2*, *Efna5*, *Mast4*) were PCR-amplified and deep-sequenced at >5000× coverage.

For sgPten candidates, *Cnot1* and *Fndc3b* showed no indels above background. In *9230020A06Rik*, small indels were detected (~17%) but were not centered at the predicted cut site and were scattered across the amplicon, likely due to sequencing noise/technical artefact rather than Cas9 activity.

For sgTrp53 candidates, we observed a single-nucleotide variant in *Efna5* and a minor insertion in *Nedd4l*; neither alteration overlapped the Cas9 cut site (Response Fig.2F).

To rule out CRISPR-induced chromosomal rearrangements that might arise during multiplex editing, we analyzed whole-genome-sequencing (WGS) data. No inter-chromosomal translocations or large deletions were detected between the two on-target loci (Supplementary table.9).

(6) Negative-control experiment (scramble sgRNA).

Primary MOSE cells transfected with the EPI-SauriCas9-Puro vector carrying a scramble sgRNA formed a few colonies after puromycin selection, but none expanded beyond passage 3. The cultures senesced and

detached, resembling wild-type MOSE behavior (Response Fig.2G). Thus, EBNA1 expression alone is insufficient to immortalize or transform MOSE cells.

In contrast, lentiviral delivery of sgRNA pair1 targeting *Pten* and *Trp53* induced efficient editing (Response Fig.2H), yet deletion of *Pten* and *Trp53* alone did not confer tumorigenicity. Together, these results indicate that EBNA1 expression combined with *Pten/Trp53* loss is required for MOSE transformation.

Response Fig.2

- (A) Amplicon deep sequencing of 10 MEPP single-cell clones.
- (B) Proportion of clones with disruption at both loci; 40/40 single-cell clones harbored indels at *Trp53* and *Pten*.
- (C) Frameshift burden across 10 clones: Frameshift events accounted for ~100% of the detected indels.
- (D) RT-qPCR quantification of *Trp53* and *Pten* mRNA relative to *Gapdh*; statistical evaluation by multiple unpaired t-tests.
- (E) Gene-set enrichment analysis comparing MEPP with MOSE using Reactome and HALLMARK collections.
- (F) Sanger sequencing of predicted off-target amplicons for sg*Trp53* and sg*Pten*.
- (G) Representative bright-field and fluorescence micrographs of MOSE cells transfected with a scramble control plasmid.
- (H) Amplicon deep sequencing of MOSE infected with sgRNA pair1. Result showed indel frequency and

frameshift frequency of *Trp53* and *Pten*, respectively.

Q3: In case of heterogeneity of MEPP cells, it could be speculated that the development of tumor subclusters identified in primary or metastatic sites is driven by loss of *Pten* and/or *Trp53*. The analysis of indels in primary and metastatic sites would be highly informative for a precise characterization of MEPP cells as ovarian cancer model.

R3: We performed amplicon sequencing of the targeted loci in tumors from both the primary site (PS) and matched omental metastases (OM). For both *Trp53* and *Pten*, the overall indel frequencies and the fraction of frameshift mutations were indistinguishable between PS and OM (Response Fig.3A–B). The slightly lower indel rates compared with Response Fig. 2 likely reflect the presence of multiple non-tumor cell types in the sequenced samples. Taken together, these results demonstrate that metastatic lesions retain the same on-target knockout status as the primary tumors, indicating that site-specific differences in *Pten/Trp53* editing did not drive metastasis in the MEPP model.

Response Fig.3

(A) On-target indel frequencies at *Trp53* and *Pten* in primary lesions (MEPP-OV) versus omental metastases (MEPP-OM). Rates were comparable between sites.

(B) Proportions of frameshifting alleles at *Trp53* and *Pten* in MEPP-OV and MEPP-OM. Frameshift fractions were similar across sites.

Q4: Semi-quantitation of IHC of staining of MEPP (Fig.5E) should be performed to confirm the comparison with IHC staining in primary and omental metastatic tissues.

R5: Thanks for your suggestion. For each marker we scored staining intensity (0–3) × positive-cell percentage (0-100%) to yield a standard H-score on six primary-site (PS) tumors and six matched omental metastases (OM) derived from MEPP allografts. *Wt1* shows no statistically significant difference between sites. Estrogen receptor α , phosphorylated ERK, and N-cadherin are all significantly up-regulated in omental metastases versus primary tumors.

Response Fig.4 Semi-quantitative immunohistochemistry reveals site-specific up-regulation of ER α , phospho-ERK and N-cadherin in omental metastases (n=6) compared with primary lesions (n=6) formed by MEPP cells. Bars denote mean \pm s.e.m. P values were calculated with a two-tailed Student's t-test.

Q5: The use of MEPP as a drug screening platform does not seem to fully recapitulate the sensitivity of human ovarian cancer cells to therapeutic candidates. Statistical analysis of PO1 drug sensitivity is required to support the use of MEPP for drug screening. Quantification of TUNEL assay of tumors should also be performed.

R5: Thanks for your suggestion.

We have added quantitative data to substantiate the translational relevance of MEPP.

Viability of PO1 organoids was measured after 72 h exposure to FK228 or thioguanine. One-way ANOVA with Dunnett's test confirmed significant reductions in viability at $\geq 5 \mu\text{M}$ FK228 and $\geq 50 \mu\text{M}$ thioguanine (Response Fig.5A). TUNEL assay results were calculated and we found that FK228 and thioguanine treated tumor had higher percentage of TUNEL-positive nuclei than vehicle group.

Response Fig.5

(A) PO1 organoid viability measured after 72h exposure of FK228 and thioguanine.

(B) Percentage of TUNEL-positive cells in vehicle, FK228 and thioguanine treated mice.

One way ANOVA, Dunnett's multiple comparison test.

Minor comments:

- 1) The sequence of sgRNA used in the study should be provided.
- 2) Figure legends should be explained with further details
- 3) Rephrase sentences, lines 126-128 and 257-259.

Response:

Thank you for your helpful suggestions. The sgRNA sequences used in this study are now listed in Supplementary Table 8. We have expanded figure legends to provide additional methodological detail, and we have rewritten the sentences on lines 126–128 and 257–259 for improved clarity and readability.

Line 126-128:

Original: To facilitate this, a novel plasmid system, EPI-SauriCas9-dual-sgRNA plasmid, was developed for efficient dual-target sgRNA screening for *Pten* and *Trp53* knocking out in murine models (Fig.2A). The EPI plasmid vectors, in contrast to traditional plasmid transient transfected vectors, additionally containing a OriP element and express the transactivator protein Epstein-Barr nuclear antigen 1 (EBNA1).

Revised: To enable simultaneous editing, we engineered an EPI-SauriCas9 dual-sgRNA plasmid that targets *Pten* and *Trp53* in murine cells (Fig. 2A). Unlike standard transient-expression vectors, the EPI backbone incorporates the EBV OriP origin and the Epstein–Barr nuclear antigen 1 (EBNA1), allowing the plasmid to

persist episomally and maintain robust SauriCas9/sgRNA expression.

Line 257-259:

Original: Since the high frequency of PTEN loss and subsequently PI3-kinase/Akt/mTOR pathway activation show significance in prostate cancer tumor progression, especially in metastatic castration resistant prostate cancer.

Revised: PTEN loss is common in prostate cancer and potently activates the PI3K–Akt–mTOR pathway, a signaling axis closely linked to tumor progression—especially in metastatic castration-resistant prostate cancer (mCRPC).

Reviewer#2

Q1: The manuscript opens with a characterization of patient-derived ovarian cancer organoids, highlighting PTEN exon 8 mutations sustaining long-term growth of one specific organoid line, PO1. As the Introduction is constructed to highlight the previously reported limitations of organoid models and the necessity to develop a better model system, it is surprising to find Fig.1 focussed merely on organoids rather than directly starting with the development of the MEPP model. The rationale for targeting Pten and Trp53 stems from their critical roles in ovarian cancer pathogenesis and their role in previous model systems, as highlighted extensively in the Introduction. In that sense, the data presented in Fig.1 appear as Supplementary Data to the actual focus of the manuscript, which is the establishment and characterization of the MEPP model, and better streamlining of the manuscript should be considered to strengthen positive aspects of the MEPP model and remove repetitive elements in the text. At the very least, an introductory sentence clarifying the purpose of studying organoids initially would strengthen the logical flow.

R1: Thank you for the helpful suggestion. To improve coherence and keep the focus on the MEPP model, we have moved the organoid characterization (previously Fig.1) to Supplementary Fig.S1. We massively condensed the corresponding text and make this section a starting point as for the next section where we screened dual sgRNA that targeted *Pten* and *Trp53*.

Q2: Lines 95/96 state “long-term propagation achieved for only a few organoid lines (Fig.1A)”. Were there organoid lines other than PO1 that sustained long-term growth? If so, why have they not been included in the analyses in Fig.1? Are they all PTEN-deficient?

R2: Thanks for your suggestion. In addition to PO1, we obtained several patient-derived organoid lines that survived beyond passage 4. These lines did not expand sufficiently for full phenotyping, so Fig.1 focuses on PO1, the only culture that propagated >10 passages and could be profiled comprehensively. Sanger sequencing indicated exon-8 *PTEN* variants in these short-term lines as well (Response Fig.6A). Their passage histories and *PTEN* status are summarized in Supplementary Table 1.

To assess *PTEN* transcript levels, we designed RT-qPCR primers targeting the indel site and quantified expression by the $\Delta\Delta C_t$ method. Relative to IOSE-80—an immortalized, non-malignant human ovarian surface epithelial cell line—*PTEN* mRNA in the organoids was reduced by approximately two-fold (Response Fig.6B).

PTEN primer-F (5'-3'): TGAGTTCCTCAGCCGTTACCT

PTEN primer-R (5'-3'): GAGGTTTCCTCTGGTCCTGGTA

Response Fig.6

(A) Bright-field image of *PTEN*-mutant organoids that persisted to passage 2 but could not be expanded beyond passage 4 (left), and Sanger sequencing results confirming exon 8 indels in *PTEN* (right). Note: the overlapping peaks in exon 8 likely reflect cellular admixture in the organoid preparation—early passages can retain a proportion of non-neoplastic/normal tissue cells.

(B) RT-qPCR quantification of *PTEN* mRNA in *PTEN*-mutant organoids relative to IOSE-80, relative to ACTB.

Q3: The authors characterize the mutational profile of PO1 by Sanger sequencing to find *PTEN* exon 8 mutations and state that these result in deletion of *PTEN*. No expression data are shown to support this claim.

R3: Thanks for your suggestion. We quantified *PTEN* transcripts by RT-qPCR using primers overlapping the exon-8 indel. Expression was referenced to IOSE-80 (an immortalized, non-malignant ovarian surface epithelial line). *PTEN* mRNA was reduced by approximately two-fold in PO1 (Response Fig.6B).

Q4: The authors then present EPI-SauriCas9 as a novel dual-sgRNA plasmid for efficient knockout of *Pten* and *Trp53*. While innovative, the novelty of this plasmid system is not sufficiently differentiated from existing episomal and SauriCas9 systems (lines 126-133). Moreover, the data presented in Fig.2 appear supplementary to MEPP development, utilizing a murine hepatocarcinoma line (Hepa1-6) rather than ovarian models. It should be considered to move Fig.2 to the Supplementary Information.

R4: We appreciate your point. We respectfully prefer to retain Fig.2 in the main text, because the EPI-SauriCas9 platform is a core innovation that enables all downstream MEPP work and its novelty is two-fold:

1. Vector design – it is the first CRISPR plasmid that couples the OriP/EBNA1 episomal backbone (long-term, non-integrative maintenance) with the compact SauriCas9 nuclease, allowing stable dual-sgRNA expression in murine primary cells where conventional transient plasmids are rapidly lost. Notably, SauriCas9 provides SpCas9-like PAM breadth while maintaining low off-target activity (see Response Fig. 2F), thereby minimizing bystander mutations that could confound tumorigenesis readouts and compromise model fidelity. To our knowledge, this work also represents the first application of an EPI (OriP/EBNA1) backbone to murine cancer model generation, rather than its more common use solely to boost editing efficiency.

2. Dual-guide optimization workflow - Fig. 2 documents how we empirically screened three sgRNA pairs and selected the pair that has the highest indel frequency in both loci. This optimization step was indispensable for the subsequent, successful editing of primary MOSE cells.

Hepa1-6 cells were used solely as a high-transfection benchmark to evaluate the editing performance of each sgRNA pair under identical episomal conditions. Initial screening indicated that sgPair1 and sgPair2 achieved comparable editing efficiencies. We next compared these two pairs in ID8, a murine ovarian epithelial cancer model, and found that sgPair1 exhibited the highest editing efficiency at both loci (Response Fig.7). Accordingly, the same vector, promoter, and sgPair1 were subsequently applied to MOSE cells to establish the MEPP model.

Including these data in the main body makes the logic of guide selection transparent and underscores why the episomal/SauriCas9 combination was necessary. These data differentiate our plasmid from existing episomal or SauriCas9 systems and justifies the inclusion of the optimization data within the main article.

Response Fig.7 Editing frequency of different sgRNA pairs targeting Trp53 or Pten in Hepa1-6 (A) or ID8 (B).

Q5: Fig. 3 marks the true beginning of the manuscript's focus: the development and in vivo characterization of the MEPP model. The data convincingly demonstrate MEPP's ability to metastasize to the omentum in an orthotopic mouse model. scRNA-seq of primary tumours and metastatic sites provides insight into the cellular composition of the primary tumour, metastasis and tumour microenvironment. **The data are largely descriptive and the section titles seem to promise more than the data support. They should be rephrased to what is really demonstrated, e.g. distinct features of primary tumour and metastases in MEPP rather than distinct metastatic features (line 169).**

R5: Thanks for your valuable suggestion. We have revised the section title and text to align with what the data demonstrate. The narrative has been edited to describe observed composition shifts and pathway biases without overstating mechanisms.

Q6: Lines 210-212 state: "Gene Set Enrichment Analysis (GSEA) identified pathways enriched in MEPP tumors that are related to PTEN and Trp53 deletions, including MAPK activation, EMT, glycolysis, and oxidative stress (Fig.4E-F)". MEPP is a different model from ID8, as I understand it. Are they derived from the same starting cells (MOSE)? Otherwise, the difference in the pathways enriched in MEPP compared to ID8 might not/not only be attributed to the presence of Pten and p53 deletions.

R6: Thanks for your comments. MEPP and ID8 are both derived from the same starting cells (MOSE). In our study, MEPP was generated by editing primary MOSE cells using EPI-SauriCas9 vector targeting *Trp53* and *Pten*, whereas ID8 is a widely used ovarian surface epithelial cell line derived by long-term passaging of MOSE [1]. Because the two models differ in derivation and passage history, direct MEPP-ID8 comparisons can reflect factors beyond *Pten/Trp53* status.

To avoid over-interpretation, we have revised the text to base genotype-linked inferences exclusively on the MEPP vs MOSE dataset. In this comparison, GSEA shows enrichment of pathways canonically associated with *Pten/Trp53* loss (mTORC activation, E2F/MYC and checkpoint programs). We now present MEPP-ID8 analyses only as contextual comparisons (Response Fig.8).

Response Fig.8 Gene-set enrichment analysis comparing MEPP with MOSE using Reactome and HALLMARK collections.

Q7: Upregulation of N-cadherin at metastatic sites as shown in Fig. 5F is expected and not indicative of the MEPP model mimicking human ovarian cancer metastatic features, as the section title suggests. Even if found in both MEPP metastases and human ovarian cancer metastases, it represents a general marker of EMT/metastasis and does not justify the conclusion that the MEPP model specifically mimics ovarian cancer metastasis. The only data that would justify this conclusion is the data presented in **Fig. 5D, but there are also pathways that are not aligned – is the conclusion of a linear relationship justified?** Moreover, looking at Suppl. Fig.S4, not all primary tumor/metastasis pairs show the increased N-cadherin, ER and p-Erk in the omentum metastatic sites. Are the stainings shown in Fig. 4 representative?

R7: Thanks for your comments.

We acknowledge that N-cadherin upregulation is a general hallmark of EMT rather than a metastasis-specific marker in ovarian cancer. To avoid over-interpretation, we have revised the section title and narrative to emphasize site-specific differences between primary and omental lesions, rather than claiming that MEPP “mimics human metastasis.”

Regarding the cross-species transcriptomic comparison, we agree that complete pathway concordance is not expected when contrasting mouse-derived with human single-cell datasets. Our plot was not intended to imply a linear (proportional) relationship; it illustrates directional concordance for key modules (e.g., EMT and inflammatory signaling) between MEPP omental lesions and human metastases, while acknowledging pathways that diverge. We have also revised the paragraph to state a directional concordance rather than a linear relationship.

For the IHC analysis (previously Suppl.Fig.S4), we note that not all primary/metastasis pairs exhibit identical marker trends. We have increased the number of paired specimens, reported the aggregate statistics in the main figure, shown representative images, and removed Suppl.Fig.S4 to minimize inconsistencies and streamline presentation (Response Fig.9).

Response Fig.9 Semi-quantitative immunohistochemistry reveals site-specific up-regulation of ER α , phospho-ERK and N-cadherin in omental metastases compared with primary tumors (n = 12). Bars denote mean \pm s.e.m. P values were calculated with Student's t-test.

Q8: Lines 247-249 state “Comparative cluster analysis revealed moderate similarities between the infiltrating immune cells in the MEPP model and those observed in human ovarian tumors (Fig. 6G-H, Suppl. Fig. 5A-B)”. As the similarities are moderate only, the title of the section “MEPP recapitulates the human tumor microenvironment” appears overstated in relation to the presented data and should be rephrased. **The section would also benefit from enhanced cross-comparisons with human datasets and experimental validation.**

R8: Thanks for your comments. We appreciate the reviewer's point and have revised the language to align with the evidence. Specifically, the section title “MEPP recapitulates the human tumor microenvironment” has been changed to “Conserved and divergent immune-microenvironment features in MEPP relative to human ovarian tumors.”

In this section, our aim is to justify MEPP as a rapid, practical ovarian cancer model in immunocompetent mice. Accordingly, we clarified that the tumor microenvironments of MEPP-OV and MEPP-OM share key features with human disease. Single-cell RNA-seq identifies the major immune compartments—T cells and macrophages—as well as smaller populations such as neutrophils, B cells, and cDC1, enabling hypothesis-driven studies of these lineages in a tumor context. We also document commonly described macrophage and T-cell subsets, supporting the model's use for more granular investigations of subset function within the TME. While orthogonal validation by flow cytometry or multiplex immunofluorescence would be desirable, our current analysis just provides evidences to state that MEPP is suitable for immune-focused ovarian cancer research.

Q9: MEPP's utility in therapeutic screening is convincingly validated by the identification of FK228 and thioguanine, which demonstrate significant efficacy in vivo. It would be interesting to include how these drugs work. Do they act on the PTEN deficiency/PI3K activation?

R9: Thank you for the comment. To clarify mechanism, we performed bulk RNA-seq on MEPP tumors treated with FK228 or thioguanine versus vehicle (n = 3 per group) and analyzed differential expression by GSEA.

Both agents produced a convergent transcriptional program characterized by up-regulation of immune/innate signaling (e.g., inflammatory response, antigen-processing/presentation) and down-regulation of proliferation programs (E2F targets, G2M checkpoint, MYC targets, DNA replication/mitotic spindle). Nevertheless, pathways related to PI3K/AKT/mTORC1 were not enriched, indicating that the in-vivo efficacy of FK228 and thioguanine in MEPP is not mediated by direct suppression of PI3K pathway activity (Response Fig.10).

Response Fig.10 Gene set enrichment analysis of MEPP tumors treated with FK228 or thioguanine using Reactome, Gene ontology and Hallmark gene sets collections. Only gene sets with FDR $q < 0.05$ are shown; labels highlight representative pathways.

Minor points:

- It is unclear what is **IOSE in Fig. 1D**, not mentioned in the text. An ovarian cancer cell line?
- Fig. 1D, 2D: **wildtype sequence missing to compare to**
- Line 120: providing should be provided
- line 126: knocking out should be knockout
- Line 138: adding should be addition
- The significance of **Suppl. Fig.S2B** is unclear and the figure is not sufficiently described.
- Remove introductory paragraph to Fig.4 (lines 194-196), as Fig.3 already analysed metastasis in the MEPP model, no need to introduce it again.
- Line 251: Fig.S4C-H should be Fig. S5C-H

Response: Thanks for your comments.

- 1) IOSE-80 (RRID: CVCL_5546) is a non-tumorigenic human ovarian surface epithelial (OSE) cell line immortalized with SV40 large T antigen [2]. It is widely used as a pseudo-normal reference for OSE biology [3]. In this study, IOSE-80 serves as the human counterpart negative control for ovarian cancer organoids.
- 2) The wild-type reference trace is labeled "IOSE-80" in Fig. 1D. The corresponding wild-type sequences are now also provided in Fig. 2D.
- 3) Grammatical errors throughout the manuscript have been corrected.
- 4) The original sentences in the noted locations have been rewritten for clarity and precision.
- 5) Further minor grammatical issues have been addressed across the text.
- 6) Suppl. Fig. S2A lists frequently mutated genes in ovarian cancer. We selected top candidates (MUC16, CTNNB1, RB1, AHNK, RELN, HMCN) to construct EPI-SauriCas9 dual-KO plasmids in combination with sgTrp53. Suppl. Fig. S2B shows transfection and Sanger results: several guide pairs displayed low editing efficiency (Muc16 g1/g2; Ctnnb1 g1; Rb1 g1; Ahnak g1/g2; Hmcn g1/g2). For pairs with higher on-target activity, MOSE cells did not sustain long-term growth after selection. These outcomes justify the Hepa1-6 optimization step for the EPI-SauriCas9 system prior to editing MOSE cells.
- 7) The introductory paragraph has been removed.
- 8) The noted typo has been corrected.

References

1. Roby KF, Taylor CC, Sweetwood JP, Cheng Y, Pace JL, Tawfik O, et al. Development of a syngeneic mouse model for events related to ovarian cancer. *Carcinogenesis*. 2000;21(4):585-91.
2. Maines-Bandiera SL, Kruk PA, Auersperg N. Simian virus 40-transformed human ovarian surface epithelial cells escape normal growth controls but retain morphogenetic responses to extracellular matrix. *American Journal of Obstetrics & Gynecology*. 1992;167(3):729-35.
3. Choi J-H, Choi K-C, Auersperg N, Leung PCK. Gonadotropins upregulate the epidermal growth factor receptor through activation of mitogen-activated protein kinases and phosphatidylinositol-3-kinase in human ovarian surface epithelial cells. *Endocr Relat Cancer*. 2005;12(2):407-21.

Response Letter

Manuscript title: EPI-SauriCas9-Based Mouse Ovarian Cancer Models Recapitulating PTEN Deletion in Patients

We would like to thank all reviewers for your time and valuable comments.

Reviewer#1

The authors have provided convincing and detailed evidence about genome editing profile of MEPP cells, as well as primary tumors and omental metastasis. Moreover, they have further supported the use of MEPP for drug screening. I appreciate the effort invested in addressing my comments, and I am satisfied that all my concerns have been fully resolved.

R1: Thanks for your time and suggestions which made the work more intact.

Reviewer#2

The authors have made a commendable effort to answer my earlier concerns, and I believe the manuscript has been improved significantly. However, a few remaining issues should be addressed before the manuscript is suitable for publication:

Q1: Supplementary Table S1: The table does not indicate the PTEN status of the cell lines, as stated in the rebuttal letter. Including this information would be valuable for the reader and would support the manuscript's emphasis on the importance of PTEN.

R1: Thanks for your suggestion. We have added PTEN status of the indicated cell lines in Supplementary Table S1. Nevertheless, for the short-term organoids that could not be maintained, insufficient materials remained for PTEN testing and are therefore annotated as "unknown".

Organid. ID	Pathology	Passage number	PTEN status
PO1	ENOC	>10	Deficient
Org#1	OCCC	2	Unknown
Org#2	HGSOC	3	Unknown
Org#3	HGSOC	2	Unknown
Org#4	SOC	2	Unknown
Org#5	HGSOC	3	Unknown
Org#6	HGSOC	4	Unknown
OV_org_2	HGSOC	4	Deficient
OV_org_3	HGSOC	4	Deficient
OV_org_4	HGSOC	5	Deficient

Revised Supplementary Table S1

Q2: Figure 6L: Please clarify how viability was assessed. This should be described in the text or figure legend, or included in the methods section. For the PO1 viability curve with FK228, the inclusion of lower doses would strengthen the data. The presented dose of 5 μ M already results in maximum efficacy and complete loss of cell viability. Testing and presenting lower doses would help determine whether partial viability is maintained at submaximal concentrations.

R2: Thanks for your suggestion. For the drug viability test of the organoid PO1, ~5000 organoid derived cells were seeded per well in a 96-well plate and drugs were added at the indicated concentrations. Cell viability was measured 24 hours later using a CCK-3D kit (APEX BIO) according to the manufacturer's instructions. This information has been added to the method section. We further tested submaximal concentrations of FK228 on PO1 viability (Response Fig.1). FK228 at lower concentrations showed decreased efficacy while still showing

more than 50% efficacy. These data suggested that FK228 is effective in PO1 organoids and is consistent with its efficacy in the MEPP model.

Response Fig.1